
# Spectral performance analysis of the Aeolus Fabry-Pérot and Fizeau interferometers during the first years of operation

Benjamin Witschas[1], Christian Lemmerz[1], Oliver Lux[1], Uwe Marksteiner[1], Oliver Reitebuch[1], Fabian Weiler[1], Frederic Fabre[2], Alain Dabas[3], Thomas Flament[3], Dorit Huber[4], and Michael Vaughan[5]

[1]Deutsches Zentrum für Luft- und Raumfahrt e.V. (DLR), Institut für Physik der Atmosphäre, 82234 Oberpfaffenhofen, Germany
[2]Les Myriades SAS, Consultancy for Optical Systems, 31000 Toulouse, France
[3]Centre National de Recherche Meteorologique, Groupe d'etude de l'Atmosphere Meteorologique, Météo-France and CNRS, Toulouse, France
[4]DoRIT, 82256 Fürstenfeldbruck, Germany
[5]Optical & Lidar Associates OLA, Buckinghamshire, United Kingdom

**Correspondence:** Benjamin Witschas, (Benjamin.Witschas@dlr.de)

**Abstract.** In August 2018, the European Space Agency (ESA) launched the first Doppler wind lidar into space which has since then been providing continuous profiles of the horizontal line-of-sight wind component at a global scale. Aeolus data has been successfully assimilated into several NWP models and demonstrated a positive impact on the quality of the weather forecasts. In order to provide valuable input data for NWP models, a detailed characterization of the Aeolus instrumental performance as well as the realization and minimization of systematic error sources is crucial. In this paper, Aeolus interferometer spectral drifts and their potential as systematic error sources for the aerosol and wind product are investigated by means of instrument spectral registration (ISR) measurements that are performed on a weekly basis. During these measurements, the laser frequency is scanned over a range of 11 GHz in steps of 25 MHz and thus spectrally resolves the transmission curves of the Fizeau interferometer and the Fabry-Pérot interferometers (FPIs) used in Aeolus. Mathematical model functions are derived in order to analyze the measured transmission curves by means of non-linear fit procedures. The obtained fit parameters are used to draw conclusions about the Aeolus instrumental alignment and potentially ongoing drifts. The introduced instrumental functions and analysis tools may also be applied for the upcoming missions using similar spectrometers as for instance EarthCARE (ESA) which is based on the Aeolus FPI design.

## 1 Introduction

Since 22 August 2018, the first European spaceborne lidar and the first ever spaceborne Doppler wind lidar, Aeolus, developed by the ESA, has been circling on its sun-synchronous orbit at about 320 km altitude with a repeat cycle of seven days (ESA, 2008). Aeolus carries a single payload, namely the Atmospheric Laser Doppler Instrument (ALADIN) which provides profiles of the wind component along the instruments line-of-sight (LOS) direction on a global scale from ground up to about 30 km (e.g., ESA, 1999; Stoffelen et al., 2005; Reitebuch, 2012; Kanitz et al., 2019; Reitebuch et al., 2020; Straume et al., 2020). With that, the Aeolus mission is primarily aiming to improve NWP and medium-range weather forecasts (e.g., Weiss-



mann and Cardinali, 2007; Tan et al., 2007; Marseille et al., 2008; Horányi et al., 2015; Rennie et al., 2021). Especially wind profiles acquired over the Southern Hemisphere, the tropics and the oceans will contribute to closing large gaps in the availability of global wind data which represented a major deficiency in the global observing system before the launch of Aeolus (Baker et al., 2014). For the use of Aeolus observations in NWP models, a detailed characterization of the data quality as well as the minimization of systematic errors is crucial. Thus, several scientific and technical studies have been performed and published in the meanwhile, addressing the performance of ALADIN and the quality of the Aeolus data products.

Based on airborne wind lidar observations (Lux et al., 2020a; Witschas et al., 2020; Bedka et al., 2021), radiosonde data (Martin et al., 2021; Baars et al., 2020) and wind profiler measurements (Guo et al., 2021) the systematic and random errors of the Aeolus L2B wind product have been analyzed and characterized for different time periods and different geolocations. These studies verified that depending on the respective period of the mission, the respective orbit direction (ascending or descending), the respective Aeolus data processor and the respective spatial difference between Aeolus observation and reference measurement, the L2B Rayleigh-clear and Mie-cloudy winds show biases of up to several meters per second. In order to overcome and solve this problem, the identification and correction of systematic error sources was required. Weiler et al. (2021a) for instance demonstrated, that the signal detectors of Aeolus have single pixels that show anomalies regarding their dark current signal, which can lead to wind speed errors of up to $30 \ \mathrm{m\,s^{-1}}$ for several hours directly after their appearance, depending on the strength of the atmospheric signal. After implementing a new measurement procedure in order to characterize the dark current signal and a corresponding correction scheme based on this data, the impact of these hot pixels could remarkably be reduced. The corresponding correction scheme is operational in the Aeolus L1B processor since 14 June 2019. Furthermore, Rennie and Isaksen (2020), Rennie et al. (2021) and Weiler et al. (2021b) revealed that small temperature fluctuations across the $1.5 \ \mathrm{m}$ diameter primary mirror of the Aeolus telescope cause varying wind biases along the orbit of up to $8 \ \mathrm{m\,s^{-1}}$. The impact of these thermal fluctuations is successfully corrected by means of ECMWF (European Centre for Medium-Range Weather Forecasts) model-equivalent winds. The correction scheme is operational in the Aeolus processor since 20 April 2020. After having corrected these systematic errors, positive impact of Aeolus data in Observing System Experiments (OSEs) could be demonstrated by Rennie and Isaksen (2020); Rennie et al. (2021) from ground up to about $35 \ \mathrm{km}$ altitude, whereas the largest impact is found in the tropical upper troposphere. Hence, it can be seen that, in order to provide reliable and accurate wind data, the identification and correction of systematic error sources is mandatory.

In this paper, Aeolus interferometer spectral drifts and thus, potential sources for systematic errors in the wind data product, are investigated by means of instrument spectral registration (ISR) measurements that are performed on a weekly basis. During an ISR measurement, the laser frequency is scanned over a range of $11 \ \mathrm{GHz}$ and thereby resolves the entire free spectral range (FSR) of the double-edge FPIs as well as five FSRs of the Fizeau interferometer. The results of ISR measurements are usually used to perform a so-called Rayleigh-Brillouin correction (RBC) within the Aeolus L2B processor which takes the different atmospheric temperature and pressure values in different altitudes and geolocations into account in order to prevent systematic errors in the retrieved winds (Dabas et al., 2008). Additionally, by a detailed analysis of the acquired interferometer transmission curves, conclusions regarding the instrumental alignment and ongoing drifts can be drawn. In order to do so, respective mathematical model functions are derived and used in non-linear fit procedures. The tools used in





this study were developed already before the launch of Aeolus based on measurements performed with the ALADIN airborne demonstrator (A2D) (Reitebuch et al., 2009) and have been adapted accordingly. A first detailed characterization of the A2D spectrometer transmission curves is given by Witschas et al. (2012), a study where the A2D was used to prove the effect of spontaneous Rayleigh-Brillouin scattering in the atmosphere for the first time. Later, the precise characterization of the FPI

transmission curves as well as the application of accurate Rayleigh-Brillouin line shape models (Witschas, 2011a, b) allowed to derive atmospheric temperature profiles from A2D data from ground up to $15.3 \, \mathrm{km}$ with systematic deviations smaller than $2.5 \, \mathrm{K}$ (Witschas et al., 2014, 2021; Xu et al., 2021).

The paper is structured as follows. First, the ALADIN instrument is shortly introduced in Sect. 2, followed by a description of the ISR measurement mode and the data set used in this study (Sect. 3). In Sect. 4, the mathematical tools used to analyze

the measured interferometer transmission curves are introduced. Afterwards, in Sect. 5, the time series of respective instrument parameters are shown. In Sect. 6 the presented results are discussed.

## 2    Instrument description

An overview sketch of the instrumental architecture of ALADIN is given in Fig. 1. In this paper, the attention is directed to the receiver side of the system. A more detailed description of ALADIN is given in (ESA, 2008; Reitebuch et al., 2018), the laser

transmitters as well as their frequency stability in space is discussed by Lux et al. (2020a, 2021).

ALADIN is equipped with two fully redundant laser transmitters, referred to as flight models A (FM-A) and B (FM-B). They are based on frequency-tripled, diode-pumped Nd:YAG laser systems emitting at a wavelength of $354.8 \, \mathrm{nm}$ (vacuum) and are switchable by means of a flip-flop mechanism (FFM). After passing through a beam splitter (BS), a half-wave plate (HWP) used to define the polarization of the laser light, a polarizing beam splitter (PBS) used to separate transmitted and received

light and a quarter-wave plate (QWP) setting the transmitted laser light to circular polarization, the laser beam is expanded and coupled out by means of a $1.5 \, \mathrm{m}$ diameter Cassegrain telescope. A small portion of the laser radiation that is leaking through the beam splitter is further attenuated and is used as internal reference signal which allows to monitor the frequency of the outgoing laser pulses as well as to perform measurements of the frequency dependent transmission curves of the interferometers as it is for instance done during ISR measurements (see also Sect. 3). The backscattered radiation from the atmosphere and the

ground is collected by the same telescope that is used for emission (mono-static configuration) and is returned to the transmit-receive optics (TRO) where a laser chopper mechanism (LCM) is used to protect the detectors from the signal returned during laser pulse emission, after a narrowband interference filter (IF) with a width of about $1 \, \mathrm{nm}$ has blocked the broadband solar background light spectrum. Furthermore, the transmit-receive optics contain a field stop (FS) with a diameter of about $88 \, \mu\mathrm{m}$ in order to set the field of view (FOV) of the receiver to be only $18 \, \mu\mathrm{rad}$ which is needed to limit the influence of the solar

background radiation and the incidence angle on the spectrometers.

Behind the transmit-receive optics, the light is directed to the interferometers that are used to analyze the frequency shift of the backscattered light to finally derive the wind speed along the LOS direction of the laser beam. In particular, the light is first directed to the so-called Mie channel via a polarizing beam splitter block (PBSB). After increasing its diameter to $36 \, \mathrm{mm}$ by



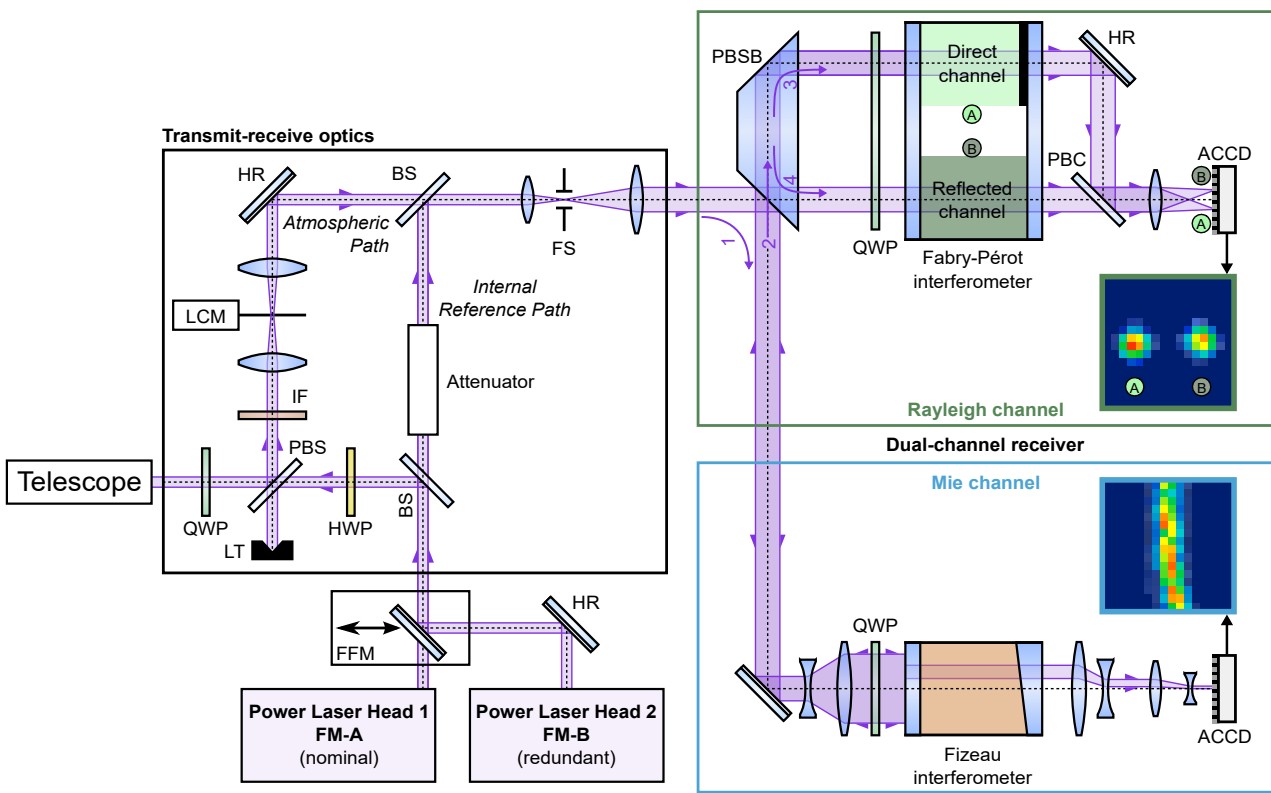

**Figure 1.** Sketch of the ALADIN optical receiver layout reproduced from Lux et al. (2021). QWP = quarter wave plate, HWP = half wave plate, PBS = polarizing beam splitter, PBSB = polarizing beam splitter block, PBC = polarizing beam combiner, FFM = flip-flop mechanism, BS = beam splitter, HR = high reflectance mirror, LCM = laser chopper mechanism, FS = field stop, IF = interference filter, LT = light trap, ACCD = accumulation charge coupled device.

means of a beam expander and with that reducing its divergence to $555~\mu\mathrm{rad}$, the light is directed to the Fizeau interferometer

which acts as a narrowband filter with a full width at half maximum (FWHM) of 58 fm (135 MHz) in order to analyze the frequency shift of the narrowband Mie backscatter from aerosol and cloud particles. The Fizeau interferometer spacer is made of Zerodur in order to benefit from its low thermal expansion coefficient. It is composed of two reflecting plates separated by 68.5 mm leading to an FSR of 0.92 fm (2190 MHz), which is chosen to be $1/5^{\mathrm{th}}$ of the FPI FSR. The plates are tilted by $4.77~\mu\mathrm{rad}$ with respect to each other and the space in-between is evacuated. The produced interference patterns (fringes) are

imaged onto the accumulation charge coupled device (ACCD) on different pixel columns, whereas different laser frequencies interfere on different lateral positions along the tilted plates. The ACCD does not image the entire spectral range covered by the aperture but only a part of 0.69 fm (1577 MHz) which is called useful spectral range (USR). This so-called fringe imaging technique using a Fizeau interferometer (McKay, 2002) was especially developed for ALADIN (ESA, 1999).

The light reflected from the Fizeau interferometer is directed towards the so-called Rayleigh channel on the same beam path

and linearly polarized in such a direction that the beam is now transmitted through the PBSB. The Rayleigh channel is based on





the double-edge technique (Chanin et al., 1989; Flesia and Korb, 1999; Gentry et al., 2000), where the transmission functions of two FPIs are spectrally placed at the points of the steepest slope on either side of the broadband Rayleigh-Brillouin spectrum originating from molecular backscattered light. For ALADIN, the two FPIs are illuminated sequentially by using the reflection of the first FPI (called direct channel or channel A) in order to illuminate the second FPI (called reflected channel or channel B). A conceptually similar approach was introduced by Irgang et al. (2002) and was adapted to the double-edge configuration for ALADIN in order to gain higher radiometric efficiency for the Rayleigh channel. This arrangement also results in different maximum intensity transmissions for both FPIs, compared to a parallel implementation of the double-edge technique with equal filter transmissions. The two FPIs are manufactured by optically contacting the plates to a fused silica spacer with a plate separation of 13.68 mm leading to an FSR of 4.6 pm (10.95 GHz), whereas the spacing of the direct channel FPI is further reduced by a deposited step of 88.7 nm (one quarter of the laser wavelength) in order to shift its center frequency with respect to the reflected channel by 2.3 pm (5.5 GHz). The space between the plates is evacuated. The plate reflectivity is measured to be 0.65 resulting in an effective FWHM of the transmission curves of 0.70 pm (1.67 GHz), whereas this value does consider defects on the plates as for instance their roughness, bowing and lack of parallelism. It does not consider any further modifications of the FWHM caused by the spectral characteristics of the light reflected from the Fizeau interferometer. This issue, and in general the shape of the interferometer transmission curves are discussed in more detail in Sect. 4.3. The light transmitted through the direct channel and reflected channel FPIs is imaged onto the same ACCD by a single lens after it was combined by an PBC with a small offset angle to $45°$, resulting in two horizontally separated circular spots. As the FPIs are illuminated with a nearly collimated beam of 1 mrad full angle divergence, only the central $0^{th}$-order of the inference pattern is imaged onto the ACCD detector. For the sake of completeness, the main specifications of the Fizeau interferometer and the FPIs are listed in table 1.

The values given above and as listed in table 1 are the essential design parameters and specifications. The actual spectroscopic performance and resultant operational parameters, as for instance the fringe width and shift, line profiles and measurement accuracies, are profoundly influenced by a multitude of optical and technical considerations. These include alignment accuracy and stability, uniformity of plate illumination, spurious and parasitic reflections, detector non-linearities, and of course any changes or fluctuations therein on short and long time scales.

The subject of this paper is thus the evaluation of the impact of these factors, based on detailed analyses of nearly three years of spaceborne data. In particular, data from ISR measurements that are performed on a weekly basis are used.

## 3   Instrument Spectral Registration (ISR)

The ALADIN instrument is able to perform special instrument modes that are used for instrument performance monitoring and calibration purposes. One of these modes is the so-called instrument spectral registration (ISR), which is used to characterize the Fizeau interferometer and the FPIs transmission curves and with that, to monitor the overall ALADIN instrumental alignment. In the following, the ISR measurement procedure as well as the corresponding data processing steps are shortly elaborated.





**Table 1.** Specifications of the Mie spectrometer and the Rayleigh spectrometer of the ALADIN instrument (Reitebuch et al., 2018).

| Mie spectrometer | Fringe imaging Fizeau interferometer |
|---|---|
| Material | Zerodur |
| Aperture | 36 mm |
| Plate spacing | 68.5 mm, vacuum gap |
| Free spectral range | 0.92 pm, 2190 MHz |
| Wedge angle | 4.77 $\mu$rad |
| Plate reflectivity | 0.85 |
| Useful spectral range | 0.69 pm, 1577 MHz |
| Fringe FWHM | 0.058 pm, 135 MHz[a] |
| Input divergence | 555 $\mu$rad full angle |

| Rayleigh spectrometer | Double-edge Fabry-Pérot interferometer |
|---|---|
| Material | Fused Silica |
| Aperture | 20 mm |
| Plate spacing | 13.68 mm, vacuum gap |
| Free spectral range | 4.6 pm, 10.95 GHz |
| Spectral spacing | 2.3 pm, 5.5 GHz |
| Plate reflectivity | 0.65 |
| FWHM | 0.75 pm, 1.78 GHz |
| Input divergence | 1 mrad full angle |

[a] Not considering a broadening induced by signal accumulation.

## 3.1 Measurement procedure

During an ISR measurement, the laser frequency is scanned over a range of 11 GHz in order to cover one FSR of the FPIs and
with that about five FSRs of the Fizeau interferometer. The data acquired during an ISR measurement contains 147 observations
and each observation itself contains 3 different frequency steps which are spectrally separated by 25 MHz. Hence, the ISR
frequency range is $(3 \times 147 - 1) \times 25\,\mathrm{MHz} = 11\,\mathrm{GHz}$. Each observation consists of 30 measurements, and each measurement
consists of 20 laser pulses, whereas the data from the last laser pulse are not acquired during the measurement. Thus, an ISR
contains the data of $147 \times (20 - 1) \times (30) = 83790$ laser pulses. These settings are not necessarily fixed but could be adapted
if required.

The raw signal measured within this procedure undergoes several preprocessing steps before being used for further investi-
gations. First, only the internal reference signal is extracted from the data product and analyzed for ISR mode measurements.
It is worth adding here that the atmospheric signal is principally available. Internal reference acquisitions with a false pulse
validity status or any other corrupt data are eliminated, but no such data was observed for the analyzed ISR data set presented
here. The remaining Rayleigh channel signal is then corrected for the detection chain offset (DCO) by subtraction of the mean
DCO level, which is implemented to avoid negative values in the digitization, as well as for the laser energy change occurring
during the laser frequency scan (see also Fig. 2). Then, the Rayleigh signal is separated into the one originating form the direct
channel (ACCD pixel 1 to 8) and the one from the reflected channel (ACCD pixel 9 to 16). The output data for the Mie channel

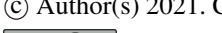


$I_{\mathrm{out}_{\mathrm{Mie}}}(f)$ and the Rayleigh channel $I_{\mathrm{out}_{\mathrm{Ray}}}(f)$ is given as the mean intensity per laser pulse according to

$$I_{\mathrm{out}_{\mathrm{Mie/Ray}}}(f) = \frac{I_{\mathrm{total}_{\mathrm{Mie/Ray}}}(f)}{N_{\mathrm{pulses}}} \tag{1}$$

where $I_{\mathrm{total}_{\mathrm{Mie/Ray}}}(f)$ is the total intensity detected per frequency step for the Mie channel or the Rayleigh channel, respectively, and $N_{\mathrm{pulses}} = 190$ is the number of laser pulses during one frequency step. Both quantities, $N_{\mathrm{pulses}}$ and $I_{\mathrm{total}_{\mathrm{Mie/Ray}}}(f)$ are reported per frequency step in a single so-called AUX-ISR auxiliary file for each ISR Aeolus data product.

## 3.2 Laser energy drift correction

As the UV output laser energy varies during the frequency scan that is executed during an ISR measurement, the intensity detected per frequency step $I_{\mathrm{total}_{\mathrm{Mie/Ray}}}(f)$ needs to be corrected accordingly. The trend of the transmitted laser energy is monitored by a photo diode (PD-74) that is mounted in the respective laser transmitter UV section (FM-A and FM-B) behind a highly reflective mirror, an additional diffuser and neutral density filters used for further signal attenuation (Lux et al., 2020b). The mean laser energy versus laser frequency derived from the ISR measurements performed between October 2018 and March 2021 is shown in Fig. 2 for the FM-A period (left) and FM-B period (right), respectively. The laser frequency is referenced to the center frequency of the direct channel FPI transmission curve (see also Fig. 7). Thus, 0 GHz indicates the center frequency of the direct channel FPI. Brownish colors correspond to the early time of the respective laser period, and blueish colors to the later times (see also the label of each panel).

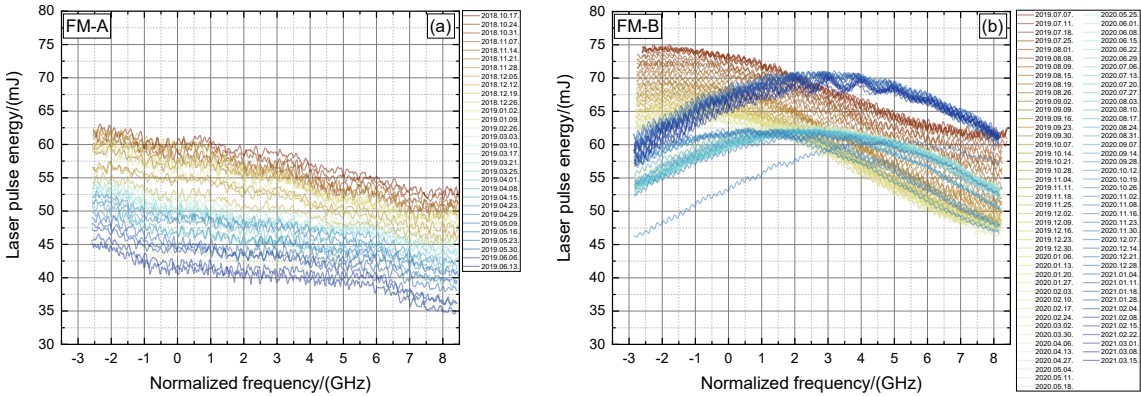

**Figure 2.** Mean laser pulse energy versus normalized laser frequency derived from ISR measurements for the FM-A period (a) and FM-B period (b).

It can be seen that the laser energy changes considerably with frequency for both lasers FM-A and FM-B. For instance, at the beginning of FM-A operation (Fig.2, left, brownish colors), the laser energy was measured to be about 62.5 mJ at lower frequencies ($\approx -2.5$ GHz) and 53.0 mJ for higher frequencies ($\approx 8.5$ GHz), which corresponds to a signal decrease of about 15% during the frequency scan. Furthermore it is obvious that, for FM-A, the laser energy is largest for lower frequencies and decreases with increasing frequency. This is also true for the early FM-B phase until February/March 2020 when a change





of the laser cold plate temperature (CPT) caused a spectral shift of the laser energy maximum to be closer to the FPI filter
cross point where also the wind measurements are performed ($\approx 2.8$ GHz). The laser cold plate couples the laser with the
laser radiator which in turn radiates the heat loss of the laser out to space. Additionally, it can be recognized that the overall
laser energy is decreasing throughout the operation time for both lasers, whereas the decrease rate is considerably larger for
the FM-A period (Lux et al., 2020b). This circumstance is discussed in more detail in Sect. 5.1. In any case, it is obvious that
the Mie and Rayleigh signals obtained during an ISR measurement need to be corrected for the varying laser energy. This is
done in the Aeolus L1B processor according to

$$I_{\text{total}_{\text{Mie/Ray}}}(f) = \frac{I_{\text{out}_{\text{Mie/Ray}}}(f)}{E_{\text{norm}}(f)} \tag{2}$$

where $I_{\text{out}_{\text{Mie/Ray}}}(f)$ is the DCO corrected raw data as given by Eq. (1) and $E_{\text{norm}}(f)$ is the normalized mean laser pulse
energy calculated according to $E_{\text{norm}}(f) = (E_{\text{PD}-74}(f))/(E_{\text{PD}-74}(f_{\text{n=1}}))$, where $E_{\text{PD}-74}(f)$ is the signal measured by PD-
74 as shown in Fig. 2, and $E_{\text{PD}-74}(f_{\text{n=1}})$ is the PD-74 measured energy at the first frequency step. Thus, $E_{\text{norm}}(f)$ is not
necessarily ranging from 0 to 1, as it is normalized arbitrarily to the value of the first data point.

A detailed analysis of the measured FPI transmission curves revealed that the energy correction of the short wave modula-
tions works reasonably well, but the correction of the overall trend seems to be insufficient. This is especially obvious from
the tilt that is visible in the relative residuals as for instance shown in the middle plot of Fig. 4, light-blue line. Such a tilt is
not explainable by incorrectnesses caused by the fit-model, as only symmetrical functions are used for the analysis (see also
Sect. 4). Thus, it is likely that the energy drift detected by PD-74 is not completely representative of the internal Rayleigh
channel signal. Hence, a modified normalized laser energy $E_{\text{norm}_{\text{new}}}(f)$ is needed for a proper energy correction. In particular,
it turned out that an additional linear correction according to $E_{\text{norm}_{\text{new}}}(n) = E_{\text{norm}}(n) + \xi \times (n/440)$ is leading to satisfying
results, with $n = 1$ to 441 being the number of data points available for ISR measurements and $\xi$ is a correction factor that
can for instance be derived by the analysis of FPI transmission curve residuals. At the beginning of the mission, this energy
correction had to be performed manually, however, since December 2018, with the implementation of the L1B processor ver-
sion 7.05, the additional energy drift correction was added. The relative residuals resulting from the additionally corrected FPI
transmission curves are represented by the blue (direct channel) and orange (reflected channel) lines in middle plot of Fig. 4. It
can be seen that the slope of the relative residuals is expectedly zero and just contains systematic deviations whose origins are
discussed in Sect. 4.3.

## 3.3 Used datasets

The first ISR measurement in space was performed on 2 Sept. 2018, only 11 days after the satellite launch. The laser was
operated at low laser pulse energies of about 11 mJ. On 8 Sept. 2018, the first ISR measurement at full laser pulse reported
energy of about 64 mJ was performed in order to verify the co-registration of the spectrometers. Co-registration means the
spectral alignment of the Mie USR center with the FPI filter cross point. After having changed the Rayleigh spectrometer
cover temperature (RCT) and having adjusted the laser frequency accordingly, the first ISR with full laser energy (59 mJ) and





co-registered spectrometers was performed on 10 October 2018. This is also the first ISR that is used in this study which ends with the ISR that was performed on 15 March 2021, the last measurement before ALADIN went to survival mode due to an instrument related anomaly. For the sake of completeness, the date, start time and mean laser energy of the ISR measurements analyzed in this study are summarized in table 2.

Table 2: Overview of the ISR data set used in this study.

| Nr. | Date[a] | Time/(UTC)[b] | Laser | Laser energy/(mJ)[c] | Nr. | Date[a] | Time/(UTC)[b] | Laser | Laser energy/(mJ)[c] |
|---|---|---|---|---|---|---|---|---|---|
| 13 | 2018-10-17 | 20-09-26 | FM-A | $57.8 \pm 3.0$ | 73 | 2019-12-23 | 03-01-59 | FM-B | $58.4 \pm 6.3$ |
| 14 | 2018-10-24 | 03-28-14 | FM-A | $57.0 \pm 3.0$ | 74 | 2019-12-30 | 03-02-11 | FM-B | $58.3 \pm 6.2$ |
| 15 | 2018-10-31 | 03-29-14 | FM-A | $56.1 \pm 3.0$ | 75 | 2020-01-06 | 03-01-59 | FM-B | $59.0 \pm 5.8$ |
| 16 | 2018-11-07 | 03-29-14 | FM-A | $54.8 \pm 2.8$ | 76 | 2020-01-13 | 03-01-59 | FM-B | $58.5 \pm 5.9$ |
| 17 | 2018-11-14 | 03-28-38 | FM-A | $55.5 \pm 3.0$ | 77 | 2020-01-20 | 03-01-59 | FM-B | $58.2 \pm 6.0$ |
| 18 | 2018-11-21 | 18-35-02 | FM-A | $55.7 \pm 4.0$ | 78 | 2020-01-27 | 03-01-59 | FM-B | $58.3 \pm 5.9$ |
| 19 | 2018-11-28 | 18-35-02 | FM-A | $52.3 \pm 3.0$ | 79 | 2020-02-03 | 03-01-47 | FM-B | $58.3 \pm 5.7$ |
| 20 | 2018-12-05 | 18-34-50 | FM-A | $51.7 \pm 3.3$ | 80 | 2020-02-10 | 03-01-47 | FM-B | $58.4 \pm 5.6$ |
| 21 | 2018-12-12 | 18-33-50 | FM-A | $50.1 \pm 2.9$ | 81 | 2020-02-17 | 03-01-59 | FM-B | $57.7 \pm 5.7$ |
| 22 | 2018-12-19 | 18-34-14 | FM-A | $55.2 \pm 4.3$ | 82 | 2020-02-24 | 03-01-35 | FM-B | $57.6 \pm 5.9$ |
| 23 | 2018-12-26 | 18-34-50 | FM-A | $54.2 \pm 3.4$ | 83 | 2020-03-02 | 03-01-47 | FM-B | $57.2 \pm 6.0$ |
| 24 | 2019-01-02 | 18-33-50 | FM-A | $54.2 \pm 3.8$ | 84 | 2020-03-30 | 03-01-47 | FM-B | $59.6 \pm 2.9$ |
| 25 | 2019-01-09 | 18-34-38 | FM-A | $52.8 \pm 3.7$ | 85 | 2020-04-06 | 03-01-35 | FM-B | $59.4 \pm 3.0$ |
| 26 | 2019-02-15 | 15-34-41 | FM-A | $42.8 \pm 2.4$ | 86 | 2020-04-13 | 03-01-35 | FM-B | $59.4 \pm 2.9$ |
| 27 | 2019-02-26 | 05-16-53 | FM-A | $41.2 \pm 3.0$ | 87 | 2020-04-20 | 03-01-47 | FM-B | $59.5 \pm 2.7$ |
| 28 | 2019-03-03 | 06-04-53 | FM-A | $44.7 \pm 3.3$ | 88 | 2020-04-27 | 03-01-59 | FM-B | $59.3 \pm 2.7$ |
| 29 | 2019-03-10 | 06-04-53 | FM-A | $49.8 \pm 2.5$ | 89 | 2020-05-04 | 03-01-47 | FM-B | $59.1 \pm 2.7$ |
| 30 | 2019-03-17 | 05-54-53 | FM-A | $48.9 \pm 2.5$ | 90 | 2020-05-11 | 03-01-47 | FM-B | $59.0 \pm 2.7$ |
| 31 | 2019-03-21 | 20-17-17 | FM-A | $48.4 \pm 2.5$ | 91 | 2020-05-18 | 03-01-47 | FM-B | $59.1 \pm 2.7$ |
| 32 | 2019-03-25 | 03-02-05 | FM-A | $48.0 \pm 2.5$ | 92 | 2020-05-25 | 03-01-47 | FM-B | $59.0 \pm 2.7$ |
| 33 | 2019-04-01 | 01-30-17 | FM-A | $46.8 \pm 2.6$ | 93 | 2020-06-01 | 03-01-35 | FM-B | $59.0 \pm 2.7$ |
| 34 | 2019-04-08 | 01-30-05 | FM-A | $45.5 \pm 2.4$ | 94 | 2020-06-08 | 03-01-23 | FM-B | $59.0 \pm 2.7$ |
| 35 | 2019-04-15 | 01-30-17 | FM-A | $44.8 \pm 2.8$ | 95 | 2020-06-15 | 03-01-35 | FM-B | $58.9 \pm 2.7$ |
| 36 | 2019-04-23 | 00-12-17 | FM-A | $47.5 \pm 2.7$ | 96 | 2020-06-22 | 03-01-35 | FM-B | $58.7 \pm 2.7$ |
| 37 | 2019-04-29 | 03-02-29 | FM-A | $46.7 \pm 2.8$ | 97 | 2020-06-29 | 03-01-35 | FM-B | $58.8 \pm 2.7$ |
| 38 | 2019-05-09 | 00-50-41 | FM-A | $43.5 \pm 1.7$ | 98 | 2020-07-06 | 03-01-35 | FM-B | $58.8 \pm 2.7$ |
| 39 | 2019-05-16 | 00-37-29 | FM-A | $43.9 \pm 2.4$ | 99 | 2020-07-13 | 03-01-23 | FM-B | $58.8 \pm 2.7$ |
| 40 | 2019-05-23 | 00-37-53 | FM-A | $43.0 \pm 2.5$ | 100 | 2020-07-20 | 03-01-23 | FM-B | $58.8 \pm 2.7$ |
| 41 | 2019-05-30 | 00-37-41 | FM-A | $42.3 \pm 2.9$ | 101 | 2020-07-27 | 03-01-35 | FM-B | $58.7 \pm 2.7$ |
| 42 | 2019-06-06 | 00-37-41 | FM-A | $40.1 \pm 2.4$ | 102 | 2020-08-03 | 03-01-23 | FM-B | $58.8 \pm 2.7$ |
| 43 | 2019-06-13 | 02-08-17 | FM-A | $39.1 \pm 2.5$ | 103 | 2020-08-10 | 03-01-35 | FM-B | $58.7 \pm 2.7$ |
| 44 | 2019-06-26 | 12-08-35 | FM-B | $3.2 \pm 0.5$ | 104 | 2020-08-17 | 03-01-35 | FM-B | $58.8 \pm 2.7$ |
| 45 | 2019-07-07 | 05-49-47 | FM-B | $68.2 \pm 4.7$ | 105 | 2020-08-24 | 03-01-35 | FM-B | $58.6 \pm 3.1$ |
| 46 | 2019-07-07 | 10-22-11 | FM-B | $68.2 \pm 4.8$ | 106 | 2020-08-31 | 03-01-23 | FM-B | $58.7 \pm 2.9$ |
| 47 | 2019-07-07 | 16-25-35 | FM-B | $68.1 \pm 4.8$ | 107 | 2020-09-07 | 03-01-35 | FM-B | $58.6 \pm 3.1$ |
| 48 | 2019-07-11 | 02-08-11 | FM-B | $67.6 \pm 4.8$ | 108 | 2020-09-14 | 03-01-23 | FM-B | $58.3 \pm 3.2$ |
| 49 | 2019-07-18 | 02-07-59 | FM-B | $67.7 \pm 4.9$ | 109 | 2020-09-28 | 03-00-59 | FM-B | $58.3 \pm 3.1$ |
| 50 | 2019-07-25 | 02-08-35 | FM-B | $66.8 \pm 5.1$ | 110 | 2020-10-12 | 03-01-59 | FM-B | $57.1 \pm 4.6$ |
| 51 | 2019-08-01 | 02-08-11 | FM-B | $66.5 \pm 5.1$ | 111 | 2020-10-19 | 03-01-23 | FM-B | $57.7 \pm 4.2$ |
| 52 | 2019-08-08 | 02-08-11 | FM-B | $65.7 \pm 5.3$ | 112 | 2020-10-26 | 03-01-11 | FM-B | $57.6 \pm 4.3$ |
| 53 | 2019-08-09 | 09-10-47 | FM-B | $65.7 \pm 5.4$ | 113 | 2020-11-02 | 03-01-23 | FM-B | $57.9 \pm 4.1$ |
| 54 | 2019-08-15 | 02-08-23 | FM-B | $64.7 \pm 5.5$ | 114 | 2020-11-08 | 02-48-11 | FM-B | $57.6 \pm 4.3$ |
| 55 | 2019-08-19 | 03-01-59 | FM-B | $64.7 \pm 5.4$ | 115 | 2020-11-16 | 03-01-11 | FM-B | $57.0 \pm 4.8$ |
| 56 | 2019-08-26 | 03-01-59 | FM-B | $63.9 \pm 5.6$ | 116 | 2020-11-23 | 03-00-59 | FM-B | $56.2 \pm 4.3$ |
| 57 | 2019-09-02 | 03-02-11 | FM-B | $63.1 \pm 5.8$ | 117 | 2020-11-30 | 03-00-35 | FM-B | $58.3 \pm 2.6$ |
| 58 | 2019-09-09 | 03-02-11 | FM-B | $62.5 \pm 6.0$ | 118 | 2020-12-07 | 03-00-23 | FM-B | $67.3 \pm 3.0$ |
| 59 | 2019-09-16 | 03-02-23 | FM-B | $61.9 \pm 6.2$ | 119 | 2020-12-14 | 03-00-35 | FM-B | $67.2 \pm 3.0$ |
| 60 | 2019-09-23 | 03-02-23 | FM-B | $61.9 \pm 6.3$ | 120 | 2020-12-21 | 03-01-11 | FM-B | $67.1 \pm 3.0$ |
| 61 | 2019-09-30 | 03-02-11 | FM-B | $61.5 \pm 6.4$ | 121 | 2020-12-28 | 03-00-59 | FM-B | $67.1 \pm 3.0$ |
| 62 | 2019-10-07 | 03-02-11 | FM-B | $61.3 \pm 6.3$ | 122 | 2021-01-04 | 03-00-59 | FM-B | $67.0 \pm 3.0$ |
| 63 | 2019-10-14 | 03-01-47 | FM-B | $61.1 \pm 6.3$ | 123 | 2021-01-11 | 03-00-47 | FM-B | $66.9 \pm 3.0$ |
| 64 | 2019-10-21 | 03-01-47 | FM-B | $60.6 \pm 6.4$ | 124 | 2021-01-18 | 03-00-59 | FM-B | $66.8 \pm 3.0$ |
| 65 | 2019-10-28 | 03-01-47 | FM-B | $60.2 \pm 6.5$ | 125 | 2021-01-28 | 20-17-59 | FM-B | $66.5 \pm 3.0$ |
| 66 | 2019-11-04 | 03-01-35 | FM-B | $59.8 \pm 6.5$ | 126 | 2021-02-04 | 20-18-47 | FM-B | $66.4 \pm 2.9$ |
| 67 | 2019-11-11 | 03-01-59 | FM-B | $59.5 \pm 6.5$ | 127 | 2021-02-08 | 03-00-59 | FM-B | $66.2 \pm 3.0$ |
| 68 | 2019-11-18 | 03-01-59 | FM-B | $59.2 \pm 6.5$ | 128 | 2021-02-15 | 03-00-59 | FM-B | $65.9 \pm 3.0$ |
| 69 | 2019-11-25 | 03-01-47 | FM-B | $59.0 \pm 6.5$ | 129 | 2021-02-22 | 03-00-47 | FM-B | $65.9 \pm 3.0$ |
| 70 | 2019-12-02 | 03-01-47 | FM-B | $59.3 \pm 6.2$ | 130 | 2021-03-01 | 03-00-35 | FM-B | $66.0 \pm 2.9$ |
| 71 | 2019-12-09 | 03-02-23 | FM-B | $58.9 \pm 6.3$ | 131 | 2021-03-08 | 03-00-35 | FM-B | $65.5 \pm 3.1$ |
| 72 | 2019-12-16 | 03-02-11 | FM-B | $58.0 \pm 6.4$ | 132 | 2021-03-15 | 03-00-47 | FM-B | $65.4 \pm 3.0$ |

[a] Date format is year-month-day.
[b] Start time of the ISR measurement, formated in hours-minutes-seconds.
[c] Mean laser energy and standard deviation per ISR measurement.



## 4   Analysis of ISR data

As explained in Sect. 3, ISR data yields the transmitted signal intensity through the Fizeau interferometer and the FPIs over a frequency range of 11 GHz. This data provides valuable information about the co-registration of the interferometers but also on the overall alignment conditions of the optical receiver as the spectral shape of the interferometer transmission curves depends on various parameters. Such parameters are the interferometer properties themselves (e.g. plate spacing, plate reflectivity, index of refraction of the medium between the plates, plate surface quality), the spectral characteristics of the laser beam (e.g. diameter, divergence, intensity distribution) and the incidence angle of the laser beam onto the interferometers. Thus, the measurement of the interferometer transmission curves and the careful analysis with respective mathematical model functions allows to investigate potential changes and drifts of the aforementioned quantities. In the following, the model functions for analyzing the interferometer transmission curves are introduced for both the Rayleigh channel (double-edge FPIs) and the Mie channel (Fizeau interferometer).

### 4.1   Fabry-Pérot interferometers

The particular characteristics of FPIs as well as the corresponding mathematical descriptions are comprehensively summarized in the textbooks by Vaughan (1989) and Hernandez (1986). Another illustrative mathematical description of the characteristics of an FPI that is applied in a direct detection wind lidar is given by McGill et al. (1997). In this section, the models used to analyze the double-edge FPI transmission curves are demonstrated and corresponding parameters describing the overall alignment conditions of the ALADIN optical receiver are introduced. It will be shown that the sequential arrangement of the interferometers requires some special treatment. Parts of the model functions have already been developed before the launch of Aeolus based on particular measurements performed with the A2D (Witschas, 2011c; Witschas et al., 2012, 2014) and were adapted to ALADIN.

The transmission function $\mathcal{T}_{\mathrm{ideal}}(M)$ of an ideal FPI (i.e., axially parallel beam of rays, mirrors perfectly parallel to each other, mirrors of infinite size, and mirrors without any defects) is described by the normalized Airy function according to

$$\mathcal{T}_{\mathrm{ideal}}(M) = \left(1 - \frac{\mathcal{A}}{1-R}\right)^2 \left(\frac{1-R}{1+R}\right) \left(1 + 2\sum_{k=1}^{\infty} R^k \cos\left(2\pi k M\right)\right) \tag{3}$$

where $\mathcal{A}$ accounts for any absorptive or scattering losses in or on the interferometer plates, $R$ is the mean plate reflectivity, and $M$ is the order of interference which can physically be considered as the number of half-waves between the interferometer plates and which can be written as

$$M = \frac{2n}{c} d f \cos(\theta) \tag{4}$$

where $f$ is the frequency of the transmitted light, $n$ is the index of refraction of the medium between the plates, $c$ is the velocity of light in vacuum, $d$ is the plate separation and $\theta$ is the incidence angle of the illuminating beam. Furthermore, the frequency





change that is needed to change $M$ by one is defined as the FSR of the interferometer $\mathcal{F}_{\mathrm{FSR}}$ and is given by

$$\mathcal{F}_{\mathrm{FSR}} = \frac{c}{2\,n\,d\,\cos(\theta)} \tag{5}$$

Additionally, the full width at half maximum $\Delta f_{\mathrm{FWHM}}$ of $\mathcal{T}_{\mathrm{ideal}}(f)$ can be calculated according to

$$\Delta f_{\mathrm{FWHM}} = \mathcal{F}_{\mathrm{FSR}} \cdot \arcsin\left((1-R)/(\pi \cdot \sqrt{R})\right) \approx \mathcal{F}_{\mathrm{FSR}} \cdot \left((1-R)/(\pi \cdot \sqrt{R})\right) \tag{6}$$

where the approximation is valid if the argument of the inverse sine has small values, which is true in case of $R$ being close to unity.

In reality, however, imperfections and irregularities on the FPI mirror's surfaces cause a change in the intensity transmission of the FPI, which has to be considered when deriving appropriate model functions. Such deviations can for instance be caused by microscopic imperfections on the mirrors, errors in their parallel alignment, or non-uniformities in the reflective coatings which cause the effective mirror separation to vary across the face of the interferometer. As for instance shown by Vaughan (1989), different defect functions can be applied to the Airy function in order to deal with the various kinds of defects. In case

of ALADIN it turned out that a normally distributed Gaussian defect function according to (Witschas, 2011c)

$$\mathcal{D}_g(f) = \frac{1}{\sqrt{2\pi}\sigma_g} \exp\left(\frac{f^2}{2\sigma_g{}^2}\right) \tag{7}$$

is well suited for that purpose. Here, $\sigma_g$ is the standard deviation of the Gaussian defect function and is called defect parameter. The convolution of Eq. (3) and Eq. (7) leads to a modified FPI transmission function normalized to unit area according to

$$\mathcal{T}_{\mathrm{Gauss}}(f) = \frac{1}{\mathcal{F}_{\mathrm{FSR}}} \left(1 + 2\sum_{k=1}^{\infty} R^k \cos\left(\frac{2\pi k}{\mathcal{F}_{\mathrm{FSR}}}(f-f_0)\right) \exp\left(-2\left(\frac{\pi k \sigma_g}{\mathcal{F}_{\mathrm{FSR}}}\right)^2\right)\right) \tag{8}$$

where $f_0$ denotes the center frequency. The effect of absorptive or scattering losses is neglected here. In case of ALADIN, also the sequential arrangement of the interferometers needs to be taken into account (see also Fig. 1), which on the one hand means that the photons within the receiver are recycled, but on the other hand means that any spectral imprint of the light reflected from one interferometer is also affecting the spectral characteristics of the transmitted light of the following interferometers. Hence, for the direct channel FPI, the spectral characteristics of the light reflected from the Fizeau interferometer have to be

considered. Accordingly, for the reflected channel FPI the spectral characteristics of the light reflected from the direct channel FPI have to be considered. Thus, the spectral shape of the light transmitted through the direct channel FPI $\mathcal{T}_{\mathrm{dir}}(f)$ is described according to

$$\mathcal{T}_{\mathrm{dir}}(f) = \mathcal{I}_{\mathrm{dir}} \cdot \left(1 + 2\sum_{k=1}^{\infty} R_{\mathrm{dir}}{}^k \cos\left(\frac{2\pi k}{\mathcal{F}_{\mathrm{FSR}_{\mathrm{dir}}}}(f-f_{0_{\mathrm{dir}}})\right) \exp\left(-2\left(\frac{\pi k \sigma_{g_{\mathrm{dir}}}}{\mathcal{F}_{\mathrm{FSR}_{\mathrm{dir}}}}\right)^2\right)\right) \cdot \mathcal{R}_{\mathrm{Fiz}}(f) \tag{9}$$





where $\mathcal{I}_{\mathrm{dir}}$ is the mean intensity per FSR, and $\mathcal{R}_{\mathrm{Fiz}}(f)$ depicts the reflection on the Fizeau interferometer which is described

by an empirically derived formula according to

$$\mathcal{R}_{\mathrm{Fiz}}(f) = \left(1 - \mathcal{I}_{\mathrm{Fiz}}\left(\cos\left(\frac{\pi}{\mathcal{F}_{\mathrm{FSR}_{\mathrm{Fiz}}}}(f - f_{0_{\mathrm{Fiz}}})\right)^4 - d_{\mathrm{Fiz}}\right)\right) \tag{10}$$

where $\mathcal{I}_{\mathrm{Fiz}}$ is the modulation depth (peak to peak), $\mathcal{F}_{\mathrm{FSR}_{\mathrm{Fiz}}}$ is the FSR of the Fizeau interferometer, $f_{0_{\mathrm{Fiz}}}$ is the center frequency (valley of the cosine function), and $d_{\mathrm{Fiz}}$ is the y-axes shift from zero and is set to be constant ($d_{\mathrm{Fiz}} = 0.5$). Although Eq. (10) is only an approximation of the complex and varying reflection function of the Fizeau interferometer, it provides suf-

ficient accuracy. This is demonstrated in Fig. 3, which shows the normalized Fizeau reflection depending on the commanded laser frequency obtained from the ISR measurement performed on 10 October 2018 (black dots) and the corresponding least-squares best-fit using Eq (10) (light blue line). In particular, what is contained in the AUX-ISR auxiliary file is the signal transmitted through the Fizeau interferometer depending on the commanded laser frequency $\mathcal{T}_{\mathrm{Fiz}}(f)$. Based on that, the reflected signal is calculated without considering any absorption or scattering losses with $\mathcal{R}_{\mathrm{Fiz}}(f) = 1 - \mathcal{T}_{\mathrm{Fiz}}(f)$. The overall

spectral shape of the Fizeau reflection is well represented by the fit in spectral regions where measurement data is available. In regions where the Mie fringe is out of the USR and not imaged onto the ACCD (e.g., 2.5 GHz to 3.0 GHz), no comparison can be performed.

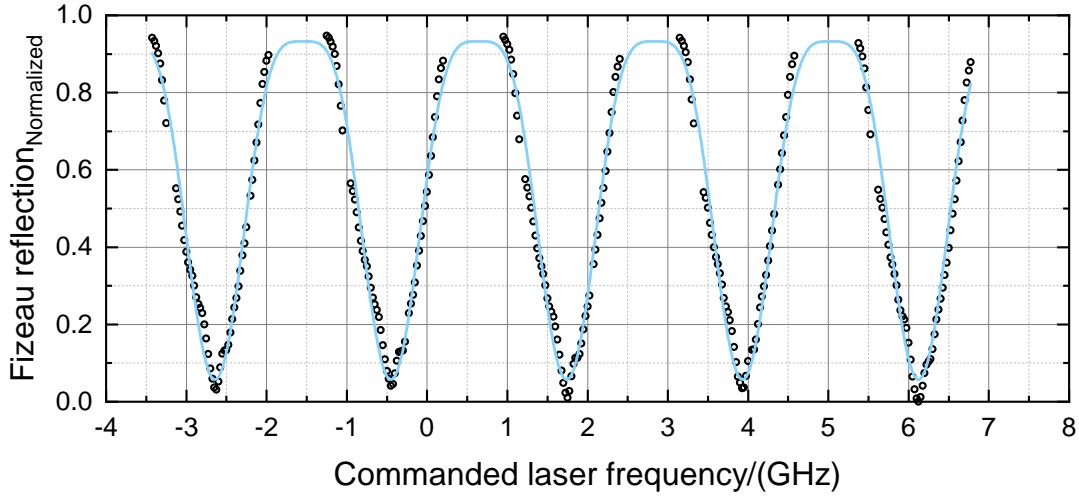

**Figure 3.** Normalized Fizeau reflection depending on commanded laser frequency obtained from an ISR measurement performed on 10 October 2018 (black dots) and the corresponding best-fit of Eq (10) (light blue line).

In order to describe the transmission through the reflected channel FPI one additionally has to consider the reflection on the direct channel FPI and furthermore a potentially leaking beam splitter (see also PBSB in Fig. 1) that could partly lead to

a direct illumination of the reflected channel FPI. Considering that, the transmission through the reflected channel $\mathcal{T}_{\mathrm{ref}}(f)$ is





described according to

$$\mathcal{T}_{\mathrm{ref}}(f) = \mathcal{I}_{\mathrm{ref}} \cdot \left(1 - Q\overline{\mathcal{T}_{\mathrm{dir}}}\right) \left(1 + 2\sum_{k=1}^{\infty} R_{\mathrm{ref}}{}^k \cos\left(\frac{2\pi k}{\Gamma_{\mathrm{FSR}_{\mathrm{ref}}}}(f - f_{0_{\mathrm{ref}}})\right) \exp\left(-2\left(\frac{\pi k \sigma_{g_{\mathrm{ref}}}}{\Gamma_{\mathrm{FSR}_{\mathrm{ref}}}}\right)^2\right)\right) \cdot \mathcal{R}_{\mathrm{Fiz}}(f) \tag{11}$$

where $\overline{\mathcal{T}_{\mathrm{dir}}} = \mathcal{T}_{\mathrm{dir}}(f)/\mathcal{T}_{\mathrm{dir}}(f_{0_{\mathrm{dir}}})$ is the normalized transmission function of the direct channel FPI. $Q$ takes into account a potentially leaking polarizing beam splitter, allowing for a $\mathcal{T}_{\mathrm{ref}}(f_{0_{\mathrm{dir}}})$ different from zero, with zero being the value of the ideal

case. All other parameters are for the reflected channel as described for the direct channel in Eq. (9).

In order to investigate the ALADIN instrumental alignment and ongoing spectral drifts, a fit of Eq. (9) and Eq. (11) to the ISR measurement data is performed by using a downhill simplex optimization method implemented in OriginLab. The sum of the Fourier series describing the Airy function is calculated for 51 terms. Considering a mean plate reflectivity of $0.65$, the neglected terms only contribute to about $0.65^{51} = 3 \cdot 10^{-10}$. Except for $\mathcal{F}_{\mathrm{FSR}} = 10946\,\mathrm{MHz}$ and $d_{\mathrm{Fiz}} = 0.5$, all parameters are

not constrained and thus a result of the fit routine. The assumption of a constant FSR is justified by the solid arrangement of the FPIs and the temperature stabilization of down to $10\,\mathrm{mK}$ which results in a rather constant plate spacing. Even alignment changes that may alter the incidence angle on the FPI by for instance $5\,\mathrm{mrad}$ would change the FSR by only $0.1\,\mathrm{MHz}$.

Based on the determined fit parameters, further quantities that characterize the FPI transmission curves can be derived. The FWHM of an ideal FPI was already introduced by Eq. (6). After introducing a defect parameter that takes into account

any imperfections and irregularities on the FPI mirror's surfaces, the FWHM can be calculated by describing the convolution of an Airy function and a Gaussian function by a Voigt function whose FWHM can be accurately approximated (Vaughan, 1989; Olivero and Longbothum, 1977). Without considering the reflection on the Fizeau interferometer, the total FWHM $(\Delta \mathrm{f}_{\mathrm{FWHM}_{\mathrm{tot}}})$ can be approximated according to

$$\Delta \mathrm{f}_{\mathrm{FWHM}_{\mathrm{tot}}} = 0.53431\,\Delta \mathrm{f}_{\mathrm{FWHM}_{\mathrm{ref}}} + \sqrt{0.21686\,\Delta \mathrm{f}_{\mathrm{FWHM}_{\mathrm{ref}}}{}^2 + \Delta \mathrm{f}_{\mathrm{FWHM}_{\mathrm{def}}}{}^2} \tag{12}$$

where $\Delta \mathrm{f}_{\mathrm{FWHM}_{\mathrm{ref}}}$ is given by Eq. (6), and $\Delta \mathrm{f}_{\mathrm{FWHM}_{\mathrm{def}}} = 2\sqrt{2\ln 2}\,\sigma_g$, which accounts for the broadening by defects. Thus, $\Delta \mathrm{f}_{\mathrm{FWHM}_{\mathrm{tot}}}$ provides a good possibility to monitor the characteristic FPI transmissions without being influenced by the Fizeau interferometer.

## 4.2  Fizeau interferometer

In a Fizeau interferometer the plates are set with a wedge angle and spacing chosen to match the spectroscopic problem. The

resultant fringes are thus localized at the plates rather than at infinity as in the FPIs. Furthermore, the fringes are straight lines rather than circular, and aligned parallel to the wedge vertex. A text book analysis of the particular characteristics of Fizeau interferometer is given by Born and Wolf (1980), drawing on the analyses of Brossel (1947) and has since been extended by many authors (e.g., Kajava et al., 1994; McKay, 2002). These calculations all essentially use classical ray optic techniques and typically show asymmetric fringes often with appreciable fringe satellites, particularly for larger wedge angles. In this simple

ray optic treatment no allowance is made for the local slope of the plates and no account is taken of diffraction effects.





For the Aeolus Fizeau interferometer the wedge angle is rather small (4.77 $\mu$rad) and the plates themselves are subject to "fine grain" surface defects of regular spiral character, due to the magnetorheological optical finishing process. In this situation the resultant fringes can only be modelled by rigorous wave optic techniques (e.g., Jakeman and Ridley, 2006; Vaughan and Ridley, 2013). The resultant wave optic fringe profiles show some asymmetries, but under the Aeolus conditions of operation

(small wedge angle and surface defects less than about $\pm 1$ nm) these were shown to be relatively small and the fringes could be sufficiently described by a Lorentzian function. Hence, for a respective ISR data set, the fringes originating at each frequency step are analyzed by fitting a Lorentzian curve according to

$$\mathcal{T}_{\mathrm{Fiz}}(x) = \mathcal{I}_{\mathrm{Peak_{height}}} \left( \frac{\Delta \mathrm{f}_{\mathrm{FWHM_{Fiz}}}{}^2}{4(x - x_0)^2 + \Delta \mathrm{f}_{\mathrm{FWHM_{Fiz}}}{}^2} \right) \qquad (13)$$

where $\mathcal{I}_{\mathrm{Peak_{height}}}$ is the peak amplitude, $\Delta \mathrm{f}_{\mathrm{FWHM_{Fiz}}}$ the FWHM and $x_0$ the position of the fringe and which is usually called

Mie response. The pixels of the ACCD are numbered from 1 to 16. Thus, when the Fizeau fringe is centered on the ACCD, the center position is half way between pixel (px) 8 and 9, namely 8.5 px. Hence, each pixel index value from 1 to 16 denotes the center of each ACCD column, resulting in a start value of 0.5 px and an end value of 16.5 px. The fit itself is performed by applying a downhill simplex optimization procedure with Eq. (13). Compared to the FPI analysis, the analysis of the Fizeau fringes is already performed by the Aeolus L1B processor. Thus, $\mathcal{I}_{\mathrm{Peak_{height}}}$, $\Delta \mathrm{f}_{\mathrm{FWHM_{Fiz}}}$ and the Mie response $x_0$ are available

in the AUX-ISR product files.

Furthermore, the Fizeau transmission is calculated at each frequency step as the intensity sum of all 16 pixel after DCO correction. Additionally, the Fizeau transmission is corrected for the laser energy change occurring during the frequency scan similar to the Rayleigh channel signals (see also Eq. (2)), however, an additional energy drift as described in Sect. 3.2 and as it is applied for the Rayleigh signal is not considered for the Mie signal. The evaluation of the Fizeau transmission, hence, gives

an approximation of potential changes in the beam intensity profile or beam diameter in one dimension.

### 4.3 Instrument functions for the Fizeau interferometer and the FPIs on 10 October 2018

The first ISR with full laser energy (59 mJ) and co-registered spectrometers was performed on 10 October 2018. The FPI transmission curves measured on that day including model fits according to Eq. (9) and Eq. (11), the corresponding relative residuals as well as the derived Mie response are shown in Fig. 4. The corresponding fit results are summarized in table 3. The

given uncertainty of the fit values denotes the standard error derived by the fit routine.

In the top panel (a), the measured FPI transmission curve of the direct channel is indicated by blue circles and the one of the reflected channel by orange circles given in least significant bits (LSB) which represent the digitized counts for the photon flux. The corresponding best-fits are depicted by the light-blue and yellow lines. Additionally, the frequency used for wind measurements (wind mode) is indicated by the dashed magenta line. The different peak transmission of both FPIs is a result of

the sequential arrangement of the interferometers. This circumstance as well as other particular features such as the intensity dip (reflected channel, $\approx 1.2$ GHz) and the modulation caused by the reflection on the Fizeau interferometer are adequately represented by the best-fit curves. The mean intensity per FSR is determined by the fit to be 3765 LSB (direct channel) and



**Figure 4.** (a): FPI transmission curves of the direct channel (blue dots) and the reflected channel (orange dots) measured on 10 October 2018, and the corresponding best-fits according to Eq. (9) and Eq. (11) in light blue (direct channel) and yellow (reflected channel), respectively. (b): Corresponding relative residuals of the direct channel (light blue) and reflected channel (yellow). Additionally, the relative residuals of an energy drift corrected data set are shown in blue (direct channel) and orange (reflected channel). (c): Corresponding Mie response, i.e. the fringe center position on the ACCD pixel.

3209 LSB (reflected channel) leading to an intensity transmission ratio of 0.85. The respective center frequencies are derived to be $-1.236$ GHz (direct channel) and $4.220$ GHz (reflected channel), leading to a spectral spacing of $5.456$ GHz, when having

the direct channel located at lower frequencies. The corresponding spectral spacing with the reflected channel being located at





lower frequencies can be calculated by using the FSR according to $10.946\,\text{GHz} - 5.456\,\text{GHz} = 5.490\,\text{GHz}$. Thus, it is almost similar on both sides of the transmission peaks and differs by only 34 MHz.

Using the mean plate reflectivity [0.649 (direct channel)/0.653 (reflected channel)] and the defect parameter [138 MHz (direct channel)/156 MHz (reflected channel)] determined by the fit, $\Delta f_{\text{FWHM}_{\text{tot}}}$ of the respective transmission curve can be

calculated by means of Eq. (12) to be 1.591 GHz (direct channel) and 1.587 GHz (reflected channel). Thus, the FWHM of the transmission curves are almost identical when neglecting the imprint of the reflection on the Fizeau interferometer. The numerically determined FWHM from the measured transmission curves however, is 1.489 GHz (direct channel) and 1.711 GHz (reflected channel), showing that the Fizeau reflection can change the actual FWHM by several hundred MHz ($\approx 14\%$). Additionally it can be seen that the spectral crossing point of the FPIs is at a commanded frequency of about 1.75 GHz, which is

also the frequency used for wind measurements (dashed magenta line).

In the middle panel of Fig. 4 (b), the relative residual of the direct and the reflected channel as well as linear-fits to the data are depicted by the light blue and yellow line, respectively. The relative deviations vary between $-2\%$ and $4\%$ (peak-to-peak), whereas the distinct modulation is caused by an insufficient description of the spectral features of the Fizeau reflection (Eq. (10)) and modulations of the incident laser beam profile and/or the transmission over the Fizeau aperture. However,

as the Fizeau reflection cannot be measured directly with the instrument being in space, it is difficult to provide a better description without correcting other features that may have a different origin. Still, these features are only a few % in amplitude and very constant over time. Thus, the other fit parameters and their temporal trends can be considered to be reliable and not impacted. Furthermore, it can be seen that the residuals show a skewness, especially visible from the direct channel data and the corresponding line-fit (light-blue line). As it was discussed in Sect. 3.2, this is caused by an insufficient laser energy

correction that was applied in the L1B processor in the early phase of the mission. When applying the model-fits to an energy drift corrected data set, the residuals are rather flat as indicated by the blue and orange line. For the sake of completeness, the fit parameters of the energy drift corrected data set are also given in table 3 by the values in brackets. It can be seen that the energy drift correction has only a minor impact on the retrieved fit parameters which lies within the fit error. Only the obtained mean intensity per FSR shows differences of about 1% for the direct channel and 3% for the reflected channel data.

The Mie response (see also Eq. (13)) is shown in the bottom panel (c). It can be seen that about five FSRs of 2.2 GHz are measured during an ISR (see also vertical dashed black lines that indicate one FSR). The actual range that can be used for wind measurements (useful spectral range, USR) is represented by the range where the Mie response is almost linear, i.e. about $\pm 0.6$ GHz around each USR center frequency or e.g. the FPI filter cross point. The Mie USR is usually projected onto the range between pixel column 4 to 14 of the ACCD. The wind measurement is performed at a commanded frequency of

1.75 GHz, resulting in a Mie fringe being located almost in the center of the ACCD detector (pixel 9.2).

As these kind of ISR measurements have been performed on a regular weekly basis, they offer the opportunity to analyze time series of the discussed fit parameters and with that to investigate spectral drifts and a change of the alignment conditions of the instrument as discussed in Sect. 5.





**Table 3.** Fit parameters according to Eq. (9), Eq. (11) and Eq. (12). The given uncertainty of the fit values denotes the standard error derived by the fit routine.

| Parameter | Unit | Dir. ch. $\mathcal{T}_{\mathrm{dir}}(f)$ | Ref. ch. $\mathcal{T}_{\mathrm{ref}}(f)$ |
|---|---|---|---|
| $\mathcal{I}$ | LSB | $3765 \pm 3$ ($3722 \pm 4$) | $3209 \pm 2$ ($3120 \pm 2$) |
| $\mathcal{I}$ ratio (integral) | - | 0.85 (0.84) | |
| $R$ | - | $0.649 \pm 0.001$ ($0.651 \pm 0.001$) | $0.653 \pm 0.001$ ($0.652 \pm 0.001$) |
| $\sigma_g$ | MHz | $138 \pm 6$ ($147 \pm 7$) | $156 \pm 6$ ($147 \pm 7$) |
| $f_0$ | GHz | $-1.236$ ($-1.239$) | 4.220 (4.217) |
| Spacing | MHz | 5456 (5456) [a] | 5490 (5490) |
| $\Gamma_{\mathrm{FSR}}$ | MHz | 10946 (fixed) | 10946 (fixed) |
| $Q$ | - | - | $0.93 \pm 0.01$ ($0.92 \pm 0.01$) |
| $\mathcal{I}_{\mathrm{Fiz}}$ | - | $0.144 \pm 0.003$ ($0.141 \pm 0.006$) | $0.136 \pm 0.003$ ($0.141 \pm 0.004$) |
| $f_{0_{\mathrm{Fiz}}}$ | GHz | $-2.689$ ($-2.691$) | $-2.582$ ($-2.573$) |
| $\Gamma_{\mathrm{FSR_{Fiz}}}$ | MHz | $2202 \pm 8$ ($2205 \pm 6$) | $2177 \pm 6$ ($2175 \pm 3$) |
| $d_{\mathrm{Fiz}}$ | - | 0.5 (fixed) | 0.5 (fixed) |
| $\mathrm{FWHM}_{\mathrm{ref}}$ | MHz | $1519 \pm 4$ ($1508 \pm 3$) | $1496 \pm 3$ ($1500 \pm 3$) |
| $\mathrm{FWHM}_{\mathrm{def}}$ | MHz | $325 \pm 14$ ($347 \pm 16$) | $368 \pm 15$ ($345 \pm 17$) |
| $\mathrm{FWHM}_{\mathrm{tot}}$ | MHz | $1591 \pm 4$ ($1589 \pm 6$) | $1587 \pm 5$ ($1580 \pm 6$) |
| $\mathrm{Finesse}_{\mathrm{tot}}$ | - | $6.88 \pm 0.02$ ($6.89 \pm 0.02$) | $6.90 \pm 0.02$ ($6.93 \pm 0.02$) |

[a] Direct channel is at lower frequencies. The second cross point is calculated by considering the measured FSR.
The fit values in brackets denote the one retrieved from the energy drift corrected data set (see also Sect. 3.2).

## 5 Temporal evolution of the Fizeau and FPI spectral transmission curves

In this section, respective fit parameters and their temporal evolution are discussed. These are the detected mean intensity per FSR of the FPIs (Sect. 5.1), the FPI center frequencies and the corresponding spectral spacing (Sect. 5.2), the FPI FWHM (Sect. 5.3) as well as the Fizeau reflection spectral position (Sect. 5.4). Additionally, the temporal evolution of the Fizeau intensity is analyzed and discussed (Sect. 5.5). All time series show the values retrieved from all ISR measurements performed between 10 October 2018 until 15 March 2021 (see also table 2).


In order to first demonstrate the notable alignment stability of the Aeolus optical receiver as well as the reproducibility of ISR measurements, all normalized FPI transmission curves are shown in Fig. 5 for the FM-A period (a) and the FM-B period (b). In order to be able to directly compare respective measurements, the direct channel transmission curves are normalized to unit area, and the reflected channel is normalized accordingly. Furthermore, the x-axes is normalized to the direct channel center
frequency and thus marks the 0 GHz. Brownish colors correspond to the early time of the respective laser period, and blueish colors to the later times (see also the dates shown in the label of each panel).

For the FM-A period, the direct channel transmission curve is very reproducible. No distinct changes can be recognized except for slight changes originating at a normalized frequency of about $-2$ GHz which might be caused by a spectral change of the Fizeau reflection. This is different for the reflected channel transmission curves. Here, a remarkable drift of both, the





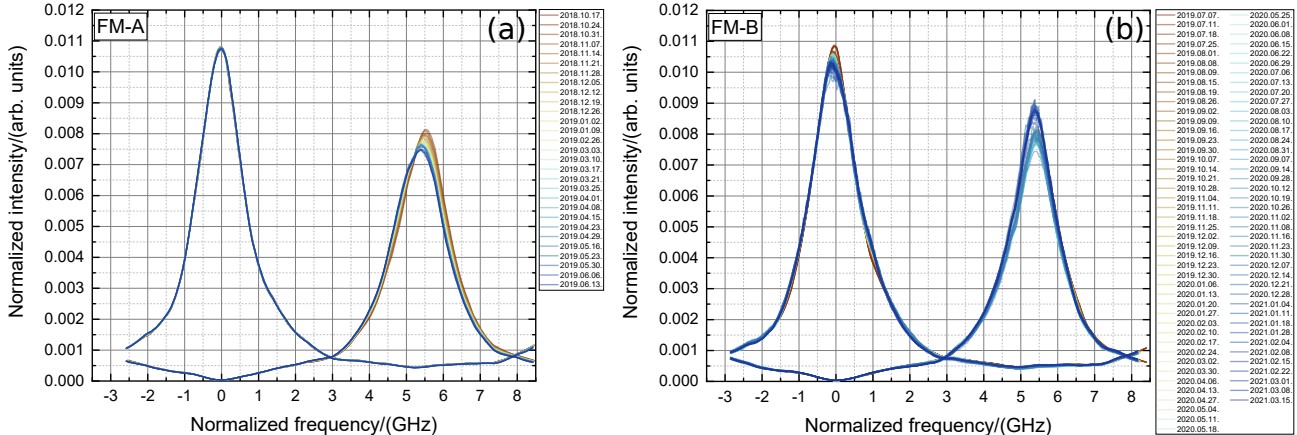

**Figure 5.** Normalized FPI transmission curves acquired during ISR measurements performed from 10 October 2018 to 15 March 2021 (see also table 2) for the FM-A period (October 2018 to June 2019, panel a) and FM-B period (July 2019 to March. 2021, panel b).

center frequency as well as the peak intensity can be observed. As the curves are normalized, this drift is with respect to the direct channel. Additionally, spectral changes are obvious and probably also caused by changing spectral characteristics of the Fizeau reflection.

For the FM-B period, a center frequency drift between the respective channels is much less or rather not observable. This will also be discussed in more detail in Sect. 5.2, which deals with the accurate analysis of the time series of the FPI center

frequencies. As for the FM-A period, the peak intensity ratio varies with time. Additionally, it can be recognized that the noise on the transmission curves starts to increase remarkably around August 2020 which is caused by enhanced signal modulations on the internal reference channel.

## 5.1   Mean intensity

One quantity that can directly be obtained from the FPI transmission curve analysis by means of Eq. (9) or rather Eq. (11)

is the mean transmitted intensity per FSR ($\mathcal{I}_{\mathrm{dir}}$ and $\mathcal{I}_{\mathrm{ref}}$) which provides information about the signal levels on the ACCD detector for the internal reference path. It is worth mentioning that $\mathcal{I}_{\mathrm{dir}}$ and $\mathcal{I}_{\mathrm{ref}}$ do not consider the impact of the reflection on the Fizeau interferometer. Thus, the real mean intensity, calculated by integrating the actual transmission curves, would slightly differ from the derived fit-values. However, an independent analysis (not shown) has verified that the obtained values are comparable and the overall trend is similar for the different analyses. The mean intensities per FSR determined for the ISR

measurements obtained from 10 October 2018 until 15 March 2021 (see also table 2) are shown in the top panel (a) of Fig. 6 for the direct channel (blue circles) and the reflected channel (orange circles), respectively. The corresponding moving averages of five successive data points are indicated by the light blue line and yellow line. The middle panel (b) shows the corresponding intensity ratio ($\mathcal{I}_{\mathrm{ref}}/\mathcal{I}_{\mathrm{dir}}$, black circles) as well as the moving average of five successive data points (gray line). In the bottom panel (c), the mean laser pulse energy during the respective ISR measurements retrieved from PD-74 is plotted. Special time





periods as for instance a instrument shut-off due to a GPS reboot anomaly (1), the switch from laser FM-A to FM-B (2) and a

laser CPT optimization period (3) are indicated by gray bars.

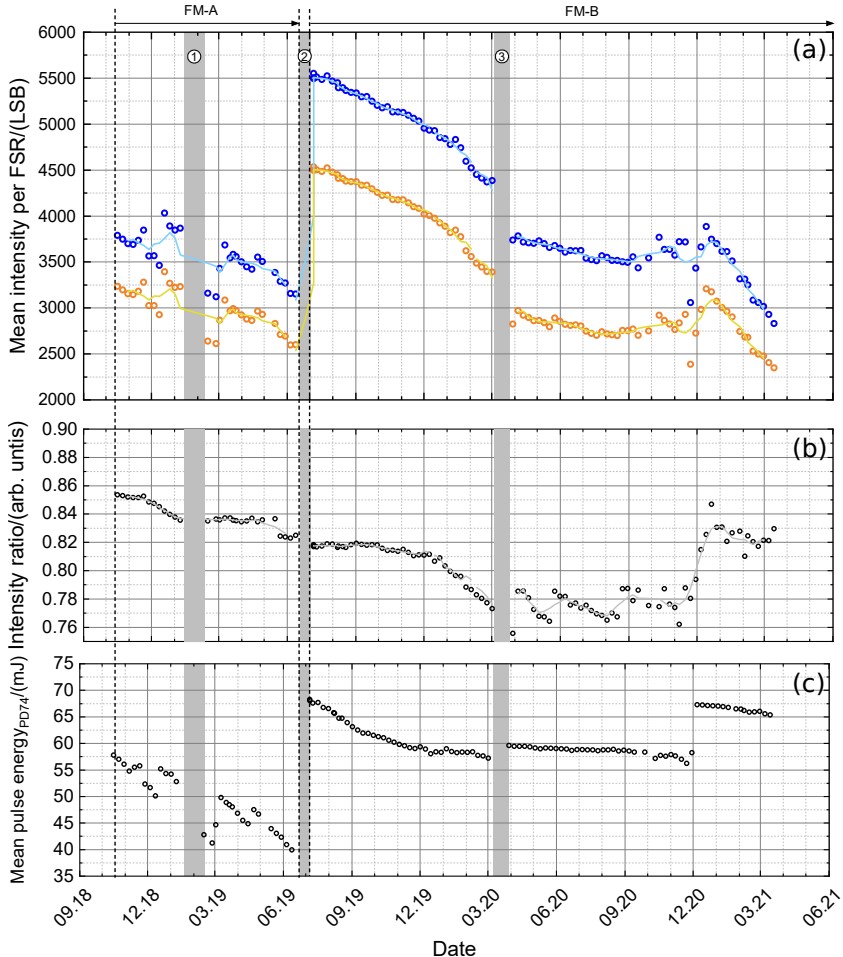

**Figure 6.** (a): Mean intensity per FSR of the direct channel (blue circles) and the reflected channel (orange circles), derived by fitting Eq. (9) and Eq. (11) to the ISR data sets obtained from 10 October 2018 until 15 March 2021 (see also table 2). The corresponding moving average of five successive data points is indicated by the light blue line and yellow line, respectively. (b): Intensity ratio ($\mathcal{I}_\mathrm{ref}/\mathcal{I}_\mathrm{dir}$, black circles) and moving average of five successive data points (gray line). (c): Mean laser pulse energy derived from PD-74. Special time periods are indicated by gray bars (1 = Instrument shut-off due to a GPS reboot anomaly, 2 = switch from FM-A to FM-B, 3 = Laser CPT optimization).

The different periods with the respective lasers FM-A and FM-B can clearly be distinguished, as the FM-A laser was

operating with about 30% lower laser pulse energy compared to FM-B (begin of life) or more precisely, the ACCD detected

30% less signal on the internal reference path. During the FM-A period, the mean intensity per FSR continuously decreased

for both channels, except for certain periods that were related to laser parameter optimizations (e.g. pre-amplifier and amplifier

current changes in mid November 2018). The decrease was almost linear in the beginning (October 2018 until November 2018

but also later on from March 2019 until June 2019), and is determined to be $(-4.7 \pm 0.5)$ LSB/d for the direct channel and





$(-4.4 \pm 0.5)$ LSB/d for the reflected channel in the time period from March 2019 to June 2019. Considering the initial intensity values of this time period of 3685 LSB (direct channel) and 3084 LSB (reflected channel), this corresponds to a decrease rate

of $0.13\%$ per day and $0.14\%$ per day, respectively. It is worth mentioning that the analysis of the PD-74 mean laser energy trend in that time period (Fig. 6, bottom) indicates a larger decrease rate of about $0.18\%$. This could for instance be explained by alignment changes that led to more photons transmitted through the internal optical path and hence, partly compensating the actual laser energy decrease. The intensity ratio (Fig. 6, middle) changed from about $0.85$ to $0.82$ during the FM-A period. This change could be due to slightly different alignment changes of the direct channel and reflected channel optical paths,

that influence the transmitted photons differently (e.g. by clipping on other optical elements) and thus, change the intensity transmission ratio or due to photons lost outside the ACCD differently for the two channels. It is interesting to realize that the trend of the intensity ratio almost continuously proceeds even when switching from FM-A to FM-B (June 2019), although the overall intensity levels remarkably increase. Furthermore from the bottom panel it can be seen that the overall mean laser pulse energy decreases from about $57.5$ mJ in October 2018 to about $40$ mJ in June 2019, which corresponds to a decrease of $30\%$

in the mentioned time period.

    Comparing the signal levels at the begin of the FM-A period (Sept. 2018) and FM-B period (July 2019), an increase of about $37.5\%$ was achieved (i.e. from 4000 LSB to 5500 LSB, for the direct channel). As for the FM-A period, the mean intensity per FSR is continuously decreasing with FM-B. In Sept. 2019, the decrease rate was $(-3.7 \pm 0.4)$ LSB/d for the direct channel and $(-2.6 \pm 0.3)$ LSB/d for the reflected channel, and thus about $40\%$ smaller than for the FM-A period. Furthermore it can

be seen that the laser CPT change that was performed in March 2020 led to a decrease of the mean intensity levels by about $15\%$ (i.e. from 4380 LSB to 3740 LSB for the direct channel), but also to a remarkable reduction of the decrease rate namely to $(-1.9 \pm 0.1)$ LSB/d for the direct channel and $(-1.5 \pm 0.2)$ LSB/d for the reflected channel. Interestingly, the mean laser pulse energy (Fig. 6, bottom) increased after the CPT changes indicating that these changes may have led to alignment changes that induced differences in the signal levels on the internal path including the Rayleigh ACCD and the PD-74. Since August 2020

the determined fit results from week to week got more variable due to appearing signal fluctuations in the internal reference signal. At the beginning of December 2020, a laser energy increase of about $15\%$ was obtained by changing laser operating parameters, however, since mid December 2020 the decrease rate increased to be $(-11.4 \pm 0.4)$ LSB/d for the direct channel and $(-9.9 \pm 0.3)$ LSB/d for the reflected channel. After the laser parameter changes in December 2020, also the intensity ratio changed rapidly (within 4 weeks) from about $0.78$ to $0.83$, which is another hint that the instrumental alignment changed

significantly within this time period. Further indications for ongoing alignment changes can be derived from the time series of the FPI center frequencies as discussed in the next section.

## 5.2   FPI center frequencies and spectral spacing

The center frequencies derived by fitting Eq. (9) and Eq. (11) to the ISR data sets obtained from 10 October 2018 until 15 March 2021 (see also table 2) are shown in the top panel (a) of Fig. 7 for the direct channel (blue circles) and the reflected

channel (orange circles), respectively, whereas the left y-axis denotes the direct channel frequencies and the right y-axis the one of the reflected channel. As the derived center frequencies show a larger jump of $5.4$ GHz between the FM-A period (Sept. 2018



until June 2019) and the FM-B period (June 2019 until now), the y-axes furthermore include a break of the same size. For a better comparability, the frequency range is chosen to be equal for both channels and both the upper and lower part of the panel, namely 150 MHz. Furthermore, line-fits to four distinct time periods with almost linear center frequency drift are shown

by light blue and yellow lines, respectively, whereas period 1 lasts from 17 October 2018 to 9 January 2019, period 2 from 15 February 2019 to 13 June 2019, period 3 from 25 July 2019 to 2 March 2020 and period 4 from 30 March 2020 to 15 March 2021. Corresponding slopes of the line fits ($\Delta f_0/\Delta t$) are given by the insets. The overall frequency changes for the direct and the reflected channel within the said time periods are 22 MHz and $-62$ MHz (period 1), 17 MHz and $-13$ MHz (period 2), 49 MHz and 34 MHz (period 3) and 47 MHz and 53 MHz (period 4). These values are additionally summarized in table 4. It

is worth mentioning that the base laser frequency of 844.961832 THz is not measured but calculated from the laser vacuum wavelength of 354.8 nm that was determined during on-ground tests (Mondin and Bravetti, 2017). The y-axes thus shows the commanded laser frequency value plus the laser base frequency. Thus the 5.4 GHz frequency jump stems from a different absolute frequency of FM-B and could be even a few FPI FSRs different.

     The bottom panel (b) of Fig. 7 shows the spectral spacing between the two FPIs which is defined as the spectral distance

between the center frequencies of the direct channel and the reflected channel according to $\Delta f_0 = f_{0_{\mathrm{dir}}} - f_{0_{\mathrm{ref}}}$, where $f_{0_{\mathrm{dir}}}$ and $f_{0_{\mathrm{ref}}}$ are the center frequencies of the direct channel and the reflected channel, respectively and $f_{0_{\mathrm{dir}}} < f_{0_{\mathrm{ref}}}$. The spectral spacing for $f_{0_{\mathrm{ref}}} < f_{0_{\mathrm{dir}}}$ can be calculated according to $\mathcal{F}_{\mathrm{FSR}} - \Delta f_0$. The gray line indicates an exponential fit to the data set of FM-A (October 2018 to June 2019) in order to visualize the settlement of the spectral spacing evolution, and the magenta line depicts a temperature time series measured at the ALADIN detection electronic units (DEU) as it is a good proxy for the

ambient temperature within the ALADIN instrument. Special time periods as for instance an instrument shut-off due to a GPS reboot anomaly (1), the switch from laser FM-A to FM-B (2) and a laser CPT optimization period (3) are indicated by gray bars.

     It can be seen that the center frequencies of both channels are drifting with time by a few MHz per week throughout the mission. This would not necessarily affect the Aeolus wind retrieval, as long as the drift would occur for both channels in

the same spectral direction and with the same rate. However, this is not inevitably true in case of ALADIN. For the FM-A period for instance, it can be recognized that $f_0$ of the respective channels was drifting in different spectral directions and with a different rate. For period 1, the center frequency drift $\Delta f_0/\Delta t$ was $(0.26 \pm 0.03)$ MHz/d for the direct channel and $(-0.73 \pm 0.03)$ MHz/d for the reflected channel. Thus, the center frequency drift of the reflected channel was about a factor of 2.8 larger and with different sign compared to the one of the direct channel. In the later FM-A phase (period 2) the

drift rate settled and equalized to $(0.14 \pm 0.01)$ MHz/d for the direct channel and $(-0.15 \pm 0.02)$ MHz/d for the reflected channel, still having a different sign. For FM-B the situation is different. Here, the center frequencies of both channels drift towards the same spectral direction (towards higher frequencies) with a comparable rate. For period 3, the center frequency drift was $(0.22 \pm 0.01)$ MHz/d (direct channel) and $(0.16 \pm 0.01)$ MHz/d (reflected channel), whereas for period 4 it was $(0.14 \pm 0.01)$ MHz/d (direct channel) and $(0.16 \pm 0.01)$ MHz/d (reflected channel). Thus, the drift rate decreased for the

direct channel but stayed constant for the reflected channel. This indicates that the overall alignment conditions or in particular





**Figure 7.** (a): Center frequencies for the direct channel (blue) and the reflected channel (orange) derived by fitting Eq. (9) and Eq. (11) to the ISR data sets obtained from October 2018 to March 2021 (see also table 2). Corresponding line fits to distinct time periods (period 1: 17 October 2018 to 9 January 2019, period 2: 15 February 2019 to 13 June 2019, period 3: 25 July 2019 to 2 March 2020, period 4: 30 March 2020 to 15 March 2021) are shown in light blue and yellow, respectively, and the derived slopes ($\Delta f_0/\Delta t$) are given by the insets. (b): Corresponding spectral spacing $\Delta f_0$ between the direct channel and the reflected channel of the FPIs (black circles) and an exponential decay fit (gray) in order to show the settlement of the spacing drift. Special time periods are indicated by gray bars (1 = Instrument shut-off due to a GPS reboot anomaly, 2 = switch from FM-A to FM-B, 3 = Laser CPT optimization).

the initial incidence angles for the internal reference channel were remarkably different for the different lasers FM-A and FM-B. In Sect. 6, these observations are used to further estimate the underlying reason for these spectral drifts.

Another quantity that can be derived from the center frequencies of both Rayleigh channels is the spectral spacing as shown in the bottom panel of Fig. 7 (black circles). The spacing is an important measure as an unconsidered spacing drift would lead



to systematic errors in the retrieved wind speeds, whereas an equal center frequency drift with similar rate and spectral drift direction (e.g. as it is almost true for the FM-B period) would not affect the wind retrieval. At the beginning of the mission, the spacing is determined to be 5450 MHz and then decreased rapidly to smaller values, which is a result of the center frequency drift occurring towards different spectral directions. Still, the drift of the spacing shows a settlement which is even independent of the switch to the FM-B laser in June 2019. Though the overall spacing changes by about 30 MHz due to the different

illumination conditions with the different lasers, the overall settlement of the drift continues. This is also confirmed by an exponential fit (Fig. 7, bottom, gray line)) applied to the FM-A data set that indicates an asymptotic convergence at 5320 MHz, which is about the spacing determined for FM-B in early 2021. This finding would allow to expect the drifting optical element being located not in the laser transmitters but between laser transmitter and optical receiver bench or for an optical element, which is common to the internal path of FM-A and FM-B if not even a rigid body motion of the laser and/or receiver optical

benches with respect to each other.

Furthermore, during the FM-B period, the spacing shows some distinct drift periods as for instance around December 2019 and around June 2020. Comparing these drifts with the ALADIN ambient temperature measured at the DEU of the system (Fig. 7, bottom, magenta line) a correlation (June 2020) and anti-correlation (December 2019 and December 2020) is obvious, whereas the temperature changes are caused by entering/leaving a solar eclipse phase where parts of the satellite orbit are in darkness leading to a temperature decrease of the instrument. This finding confirms that the ambient temperature in the

system changes and that these temperature changes have an impact on the overall alignment of the instrument, even though the temperature of the FPIs only changes by 10 mK during these eclipse periods (not shown).

### 5.3 Full width at half maximum

The FWHM of the FPI transmission curves can be calculated according to Eq. (12) using the plate reflectivity and the defect

parameter determined by the fit of Eq. (9) and Eq. (11) to the measured FPI transmission curves. The FWHM for the ISR data sets obtained from 10 October 2018 until 15 March 2021 (see also table 2) are shown in Fig. 8 for the direct channel (blue circles) and the reflected channel (orange circles), respectively.

It can be seen that the FWHM of both channels was rather similar at the beginning of the mission namely about 1590 MHz. During the FM-A period, the direct channel FWHM increased rather constantly by about 60 MHz whereas it only slightly

decreased for the reflected channel by about 20 MHz. When switching to FM-B, the obtained direct channel FWHM shows a decrease of about 70 MHz, indicating different alignment conditions for both lasers. Interestingly, the jump is smaller for the reflected channel, namely about 20 MHz. At the beginning of the FM-B phase, the direct channel FWHM shows an increase from about 1580 MHz to 1650 MHz where it settles around December 2019. Interestingly, this is the same value that was reached in the end of the FM-A period. On the other hand, the reflected channel shows again a FWHM decrease at

the beginning of the FM-B period and seems to settle in May 2020 at a value of about 1570 MHz. Since August 2020 the determined fit results from week to week got more variable due to appearing intensity variations in the degrading internal reference signal transmission (see also Sect. 6).



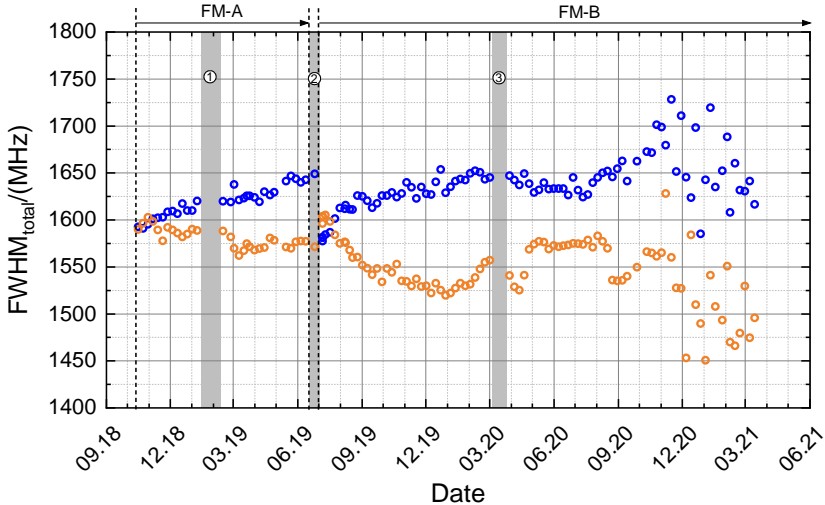

**Figure 8.** Total FWHM of the direct channel (blue circles) and the reflected channel (orange circles), derived by fitting Eq. (9) and Eq. (11) to the ISR data sets obtained from 10 October 2018 until 15 March 2021 (see also table 2) and calculated according to Eq. (12). Special time periods are indicated by gray bars (1 = Instrument shut-off due to a GPS reboot anomaly, 2 = switch from FM-A to FM-B, 3 = Laser CPT optimization).

## 5.4 Fizeau reflection spectral position

The reflection on the Fizeau interferometer has an impact on the FPI transmission curves. Hence, drifts of the spectral char-
acteristics of the Fizeau interferometer reflection can also be derived from FPI analyses. As shown with Eq. (10), the center frequency $f_{0_{\mathrm{Fiz}}}$ of the Fizeau reflection is a free fit parameter for the model functions describing the FPI transmission curves. The values determined by fitting Eq. (9) and Eq. (11) to the ISR data sets obtained from October 2018 to March 2021 (see also table 2) are shown in Fig. 9 for the direct channel (blue circles) and the reflected channel (orange circles), respectively. The corresponding moving averages of five successive data points are indicated by the light blue line and yellow line. Special time
periods as for instance a instrument shut-off due to a GPS reboot anomaly (1), the switch from laser FM-A to FM-B (2) and a laser CPT optimization period (3) are indicated by gray bars.

It can be seen that the Fizeau center frequency is continuously drifting during the FM-A period. In particular, the drift is about 100 MHz within nine months. Additionally, it can be recognized that the derived trend of the Fizeau center frequency is rather similar for the direct and the reflected channel but has an offset of about 50 MHz, which could be due to the fact that the
reflection on the Fizeau is not exactly periodical and thus, the imprint on $\mathcal{T}_{\mathrm{dir}}$ and $\mathcal{T}_{\mathrm{ref}}$ is not exactly the same. Nevertheless, it can be concluded that the spectral imprint of the Fizeau reflection on the FPIs is drifting over time during the FM-A period.

During the FM-B period, the situation is different. After switch-on of FM-B, a Fizeau center frequency drift of about 80 MHz was observed until Sept. 2019, followed by an obvious settlement. In December 2019 a drift of about 50 MHz was determined from the direct channel analysis, whereas the reflected channel showed a drift by the similar amount but in the
different spectral direction. This drift is related to a laser CPT jump that occurred on 9 December 2019. The CPT drift that





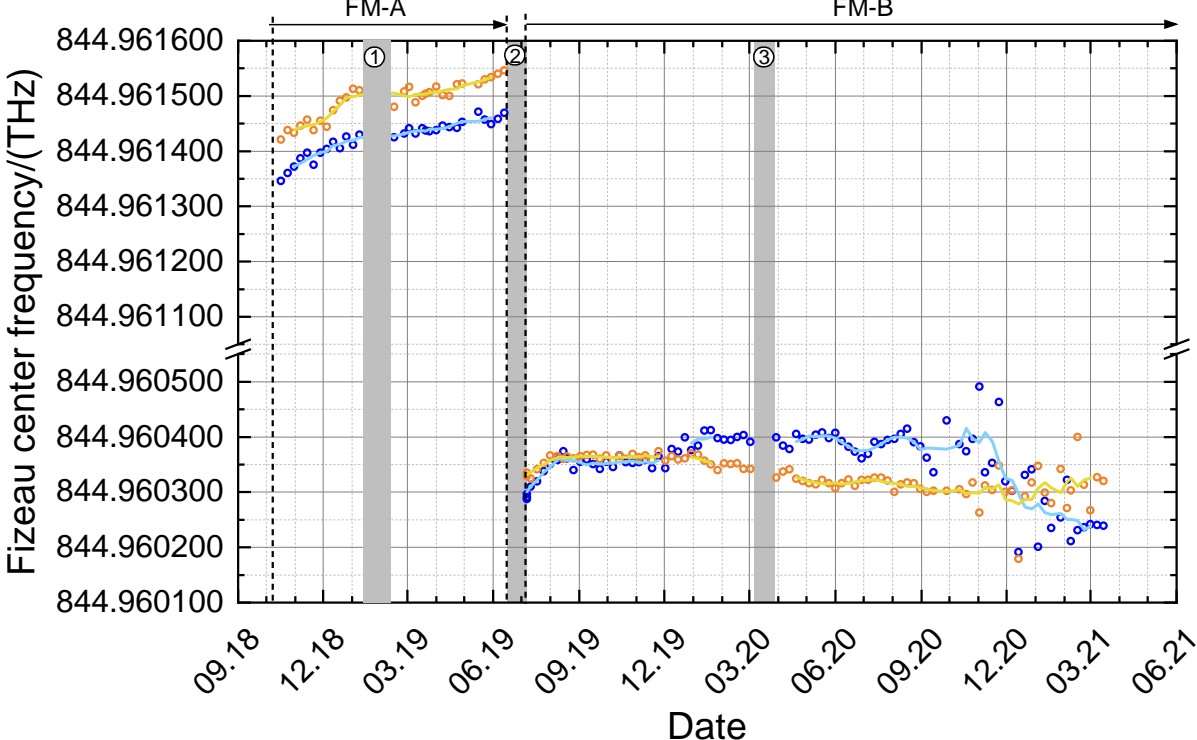

**Figure 9.** Fizeau center frequency derived from the direct channel (blue circles) and the reflected channel (orange circles) by fitting Eq. (9) and Eq. (11) to the ISR data sets obtained from October 2018 to March 2021 (see also table 2). The corresponding moving average of five successive data points is indicated by the light blue line and yellow line, respectively. Special time periods are indicated by gray bars (1 = Instrument shut-off due to a GPS reboot anomaly, 2 = switch from FM-A to FM-B, 3 = Laser CPT optimization).

happened afterwards is responsible for the Fizeau center frequency drift. With the settlement of the laser CPT, also the derived Fizeau center frequencies settled until about August 2020. Since then, the obtained fit parameters are in general more variable due to a larger variability of the spectrometer signals which could be explained by beam clipping happening due to the ongoing alignment drift (see also Fig. 7) and Fig. 6. Besides the analyses of the Fizeau reflection impact on the Rayleigh channel signals,

also Mie channel signals are available from ISR measurements for further investigations of potentially ongoing spectral drifts. The results of the Mie signal analyses are discussed in the next section.

## 5.5 Fizeau intensity

The Fizeau intensity is calculated as the sum of the laser energy and DCO corrected Mie signal at each frequency step, and thus, gives an approximation of potential changes in the beam intensity profile or beam diameter that is illuminating the Fizeau

interferometer (in one dimension). It is worth mentioning that any changes of the measured Fizeau intensity are mainly caused by a variation of the interferometer illumination of the internal path rather than due to a change of the Fizeau transmission itself, as the overall illumination is mainly determined by the laser beam profile due to the near-field image of the beam that is used





within the fringe imaging technique. The integrated Fizeau transmissions versus Mie response [Eq. (13)] obtained during ISR measurements performed from 10 October 2018 until 15 March 2021 (see also table 2) normalized to unit area (to emphasize

the beam shape change) are shown in Fig. 10 for the FM-A period (a) and the FM-B period (b). Brownish colors correspond to the early time of the respective laser period, and blueish colors to the later times (see also the label of each panel).

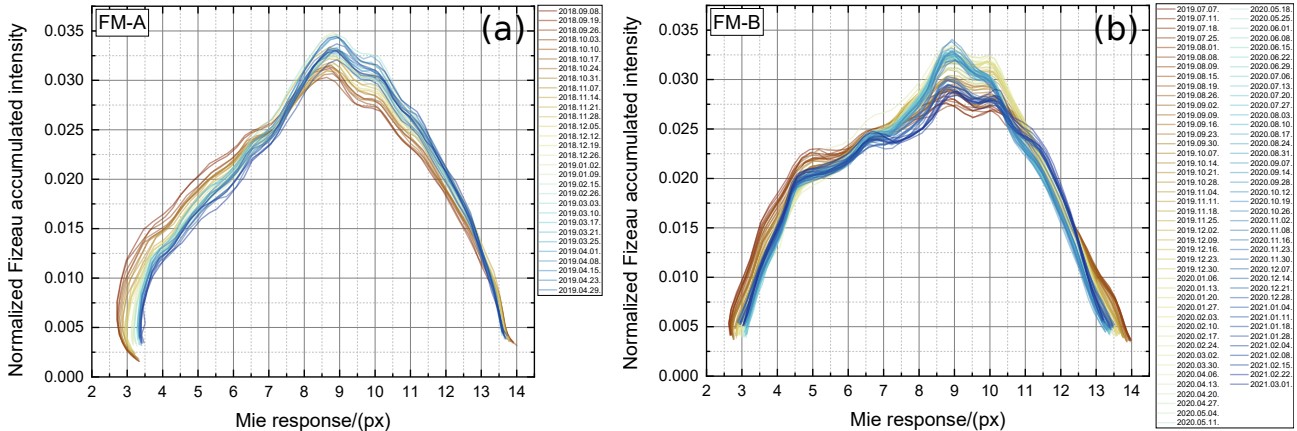

**Figure 10.** Integrated Fizeau intensity versus Mie response obtained during ISR measurements (see also table 2) normalized to unit area for the FM-A (a) and the FM-B (b).

What immediately can be seen is that the Fizeau transmission looks different for FM-A and FM-B, and that it evolves with time. Although the pronounced maximum around the Mie response of $9$ px is similar for FM-A and FM-B, the distribution for smaller Mie responses looks different, which points to a different intensity distribution of the illuminating beam or rather

different illumination conditions as for instance clipping on other optical elements. Additionally, it can be recognized that for both lasers, the width of the Fizeau transmission is decreasing with time which could be explained by a shrinking beam diameter or a change in the divergence of the illuminating beam. Furthermore, it is obvious that not only the width but also the overall spectral features are evolving which might be explained by a changing intensity distribution of the illuminating beam.

## 6  Discussion of spectral drifts

In Fig. 7 of Sect. 5.2, the time series of the determined FPI center frequencies was shown and demonstrated different drift behavior for the respective lasers FM-A and FM-B. In order to understand the observed results and in order to relate them to respective alignment changes, the equations discussed in Sect. 4.1 have to be revised as they only consider an ideal FPI with mirrors of infinite size, being illuminated normal to the optical axis of the FPI plates by a perfectly collimated beam. In reality however, the illumination cone of the light beam passing through the FPI has a certain FOV with an angular radius $\theta_\mathrm{F}$,

also called the input divergence (half of the full cone angle). For the ALADIN internal reference signal $\theta_{\mathrm{F_{INT}}}$ is assumed to be $455/2$ $\mu$rad, as determined by the parameters of the reference laser beam as it enters the FPI. In contrast, for the atmospheric signal, $\theta_{\mathrm{F_{ATM}}}$ is estimated to be $1.44/2$ mrad, as determined by a pinhole aperture in the optical chain. Furthermore, the light





beam may illuminate the FPI at a certain angle of incidence $\theta_A$. Such a situation is illustrated in Fig. 11 for $\theta_A \approx 2 \cdot \theta_F$ with

0 denoting the prime optical axis of the system defined as the normal to the FPI plates. Hence, the FPI circular interference

575  fringes (black rings, not shown for the full circle) are centered at 0, but for an off-axis aperture (gray circle) centered at $\theta_A$

only portions of the fringes will be illuminated as indicated by the orange, light-blue and dark-blue circular arcs. The dashed

lines exemplarily mark the crossing points of the second interference fringe with the aperture or rather the fraction of the

second interference fringe that is transmitted through the aperture (light blue circular arc). The asymmetric behavior of such

an off-axis illuminated FPI has been treated in detail by Hernandez (1974) as well as in the text books by Hernandez (1986)

580  and Vaughan (1989), whereas only the most important points are recapitulated here. The following analysis is based on the

principle demonstrated in Fig. 11.

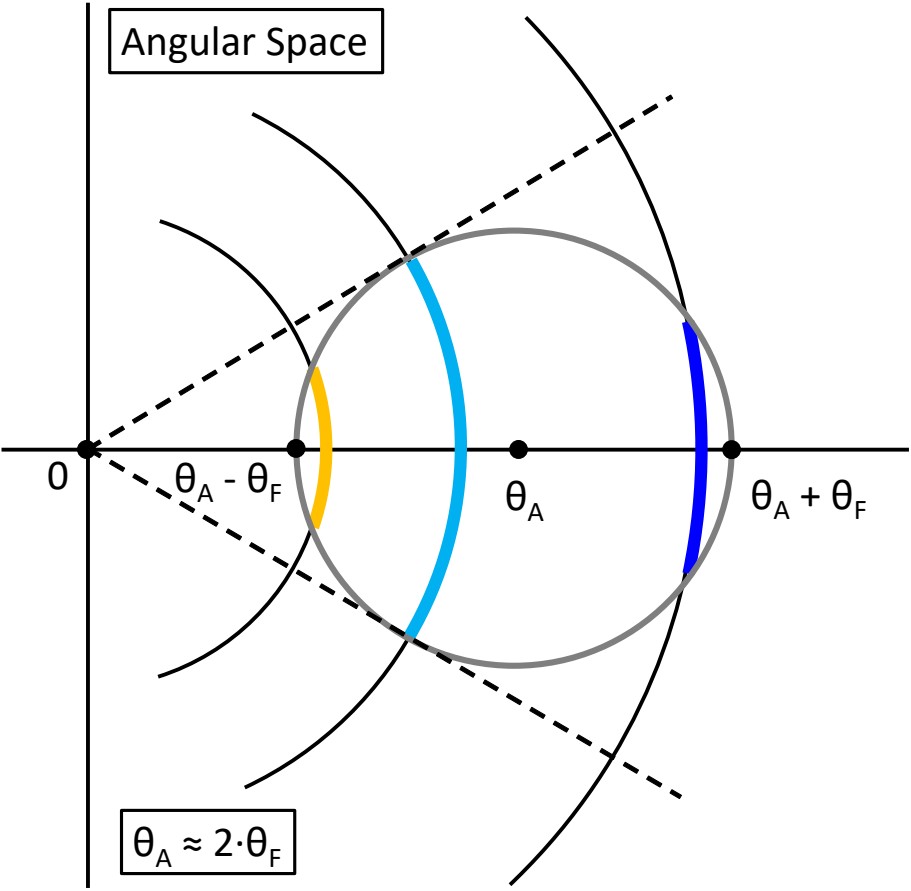

**Figure 11.** Illustration of FPI operation with an angle of incidence $\theta_A$ and a field of view $\theta_F$ in angular space with 0 denoting the prime optical axis of the system defined as the normal to the FPI plates. The FPI circular interference fringes (black rings) are centered at 0, but only portions of the fringes will be illuminated due to the off-axis centered aperture (gray circle) as indicated by the orange, light-blue and dark-blue circular arcs. The dashed lines exemplarily mark the crossing points of the second interference fringe with the aperture or rather the fraction of the second interference fringe that is transmitted through the aperture (light blue circular arc). The detailed analysis leading to Eqs. (14) to (19) and to the dispersion curves shown in Fig. 12 are based on the principle demonstrated here.





The ideal FPI, as discussed in Sect. 4.1, is an angle dependent filter with quadratic dispersion. In particular, the transmission for a narrow beam ($\theta_F \to 0$) at a certain $\theta_A$ experiences a frequency shift $\Delta f$ compared to the beam at normal incidence ($\theta_A = 0$) according to

$$\Delta f = \left(\frac{\theta_A}{\theta_1}\right)^2 \Gamma_{\mathrm{FSR}} \tag{14}$$

where $\theta_1 = \sqrt{\lambda/d} = 5.093$ mrad is the angular radius of the first full fringe and is used as a scaling reference (Vaughan, 1989) as is relates certain angular radii together with $\Gamma_{\mathrm{FSR}}$ to a frequency shift compared to the beam at normal incidence. More precisely, $\theta_1$ is the angular radius of the first fringe from the center supposing that a bright fringe of zeroth order exists exactly at the center of the pattern. Hence, the radius of the first full fringe corresponds to one FSR. From Eq. (14) it can also be seen that any change of the angle of incidence would lead to a center frequency increase as it is true for the FM-B period. For ALADIN, $\lambda = 354.8$ nm and $d = 13.68$ mm and $\Gamma_{\mathrm{FSR}} = 10.95$ GHz are given according to table 1. Additionally, by replacing $\theta_1$ and $\Gamma_{\mathrm{FSR}}$ with their definitions given above and in Eq. (5), the relative frequency shift ($\Delta f / f$) compared to the beam at normal incidence is derived to be

$$\frac{\Delta f}{f} = \frac{\theta_A^2}{2}. \tag{15}$$

For a larger beam with a non-negligible $\theta_F$, two regions of operation may be considered with $\theta_A \leqq \theta_F$ and $\theta_A > \theta_F$. For $\theta_F > 0$ and nominal incidence ($\theta_A = 0$), the aperture profile extends out to

$$\Delta f_0 = \left(\frac{\theta_F}{\theta_1}\right)^2 \Gamma_{\mathrm{FSR}} \tag{16}$$

and is a top-hat function with a full width $\Delta f_0$. The median position of this aperture function, which denotes the center of energy, is at half of this value i.e. $\Delta f_M = \Delta f_0/2$ and the peak intensity is usually designated as $I_0$. For $0 < \theta_A \leqq \theta_F$, the aperture profile starts to become asymmetric and extends out to a full width $\Delta f_W$ given by

$$\Delta f_W = \left(\frac{\theta_F + \theta_A}{\theta_1}\right)^2 \Gamma_{\mathrm{FSR}}, \tag{17}$$

however, the median position of the energy distribution remains constant up to $\theta_A \approx 0.293 \cdot \theta_F$ according to $\Delta f_M = \Delta f_0/2$. For $0.293 \cdot \theta_F < \theta_A \leqq \theta_F$, the aperture profile becomes increasingly asymmetric with a full width given by Eq. (17). However, the peak intensity remains at $I_0$. The median position $\theta_M$ is given by $\theta_M^2 = \theta_A^2 + (\kappa \theta_F)^2$, so that

$$\Delta f_M = \left(\frac{\theta_A}{\theta_1}\right)^2 \Gamma_{\mathrm{FSR}} + \kappa \cdot \left(\frac{\theta_F}{\theta_1}\right)^2 \Gamma_{\mathrm{FSR}}, \tag{18}$$





where $\kappa \approx 0.6$ to a good approximation. When $\theta_A > \theta_F$, the incident beam no longer overlaps the optical axis normal to the FPI plates and the angular position of the peak of the profile is at $\theta_p$ given by $\theta_p{}^2 = \theta_A{}^2 - \theta_F{}^2$ and in frequency terms

$$\Delta f_p = \left(\frac{\theta_p}{\theta_1}\right)^2 \Gamma_{\mathrm{FSR}} = \left[\left(\frac{\theta_A}{\theta_1}\right)^2 - \left(\frac{\theta_F}{\theta_1}\right)^2\right] \Gamma_{\mathrm{FSR}}, \tag{19}$$

and the peak intensity $I_p$ is given by

$$I_p = \frac{I_0}{\pi} \sin^{-1}\left(\frac{\theta_F}{\theta_A}\right). \tag{20}$$

However the median value which denotes the energy center of the profile is still calculated by Eq. (18) with $\kappa \approx 0.6$.

The corresponding dispersion curves of frequency shift (i.e. median position) versus angle of incidence $\theta_A$, calculated by means of Eq. (14) to Eq. (20) for beams of different angular aperture $\theta_F$ are shown in Fig. 12. The green line denotes the ideal case with $\theta_F = 0$, the red line indicates the case for the internal reference beam with $\theta_{F_{\mathrm{INT}}} = 455/2 \; \mu$rad, and the black line

depicts the case for the atmospheric signal beam with $\theta_{F_{\mathrm{ATM}}} = 1.44/2$ mrad.

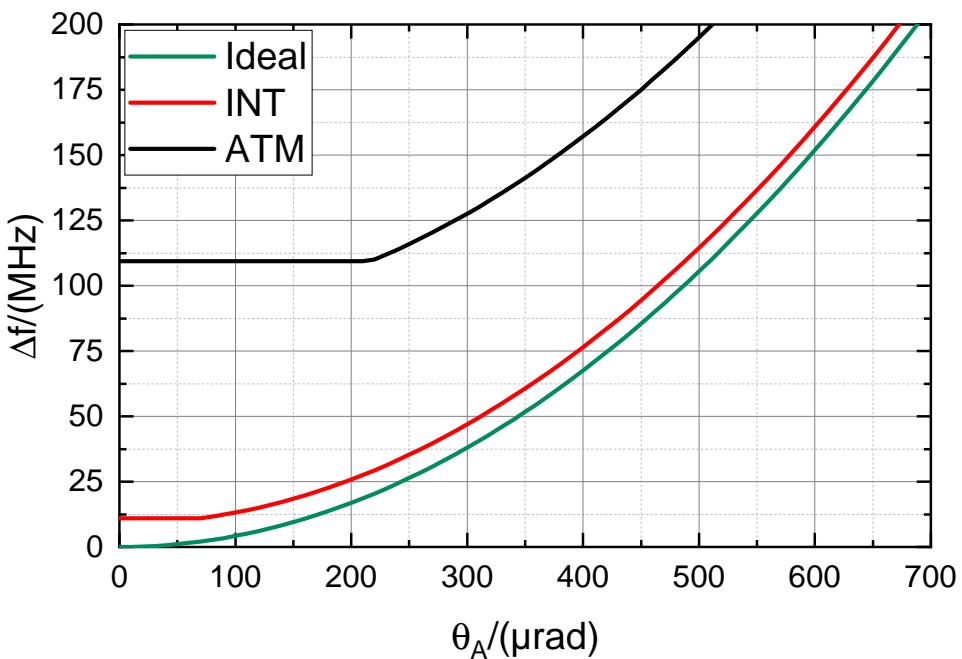

**Figure 12.** Dispersion curves of frequency shift versus angle of incidence for beams of different angular aperture (FOV) with $\theta_F = 0$ as corresponding to the ideal case (green), $\theta_{F_{\mathrm{INT}}} = 455/2 \; \mu$rad as corresponding to the internal reference beam (red) and $\theta_{F_{\mathrm{ATM}}} = 1.44/2$ mrad as corresponding to the atmospheric signal beam (black). The frequency shift between internal reference signal and atmospheric signal at nominal incidence is close to 100 MHz.





With Fig. 12 it can be seen that incidence angle drifts of a few hundred $\mu$rad are needed in order to explain the observed center frequency drifts (see also table 4) for the internal reference beam. It is further obvious that the exact incidence angle drift is depending on the initial incidence angle which is essentially unknown. During the FM-A period, the center frequency drifts by $+39$ MHz for the direct channel and by $-75$ MHz for the reflected channel. Thus, as the center frequency decreases

for the reflected channel, it can be concluded that the initial incidence angle for the reflected channel was definitely different from normal incidence. For instance, an initial incidence angle of about $\theta_{A_i} = 425$ $\mu$rad and a drift towards normal incidence could explain the observed center frequency drift. For the direct channel, the observed center frequency drift of $+39$ MHz could be explained by an incidence angle change of about $325$ $\mu$rad supposing $\theta_{A_i} = 0$. For $\theta_{A_i} = 400$ $\mu$rad, the incidence angle only has to drift by about $110$ $\mu$rad to $\theta_A = 510$ $\mu$rad in order to explain the observed center frequency drifts.

For the FM-B period, the frequency drifts of both channels are comparable and with similar sign. In particular, the center frequency drifts by $+96$ MHz for the direct channel and by $+87$ MHz for the reflected channel. Supposing an initial incidence angle of $\theta_{A_i} = 0$ an incidence angle change of about $475$ $\mu$rad is needed to explain the observed center frequency drifts. For larger initial incidence angles, the corresponding drift would accordingly be smaller. What can also be recognized during the FM-B period is that the center frequency drift rate for the direct channel is decreasing over time from $(+0.22 \pm 0.01)$ MHz/d (pe-

riod 3) to $(+0.14 \pm 0.01)$ MHz/d (period 4). As the dispersion curve gets steeper for larger $\theta_A$, this behavior can only be explained by a decreasing $\theta_A$-drift rate or a clipping of the beam which would alter the apparent $\theta_F$.

Additionally, from Fig. 12 it can be recognized that even for normal incidence ($\theta_A = 0$), the atmospheric signal is already shifted by about $100$ MHz with respect to the internal reference signal, which is due to the considerably larger FOV of the atmospheric signal and which is an important difference between the two signal channels.

Summarized, it can be said that the overall alignment conditions were significantly different for the FM-A and FM-B period. Furthermore, it has to be pointed out that a $\theta_A$-drift of several hundred $\mu$rad is rather significant and could well lead to clipping of the beam, considering the angular size of the field stop at Rayleigh spectrometer level of $1440$ $\mu$rad and an angular size of the Rayleigh spots of about $968$ $\mu$rad ($4\sigma - \mathrm{diameter}$). Thus, the larger variability of the retrieved fit parameters originating in August 2020 could be explained by larger variations induced by beam clipping. The spacing drift (Fig. 7, b) and the clear

correlation/anti-correlation to the instrument temperature confirms the temperature sensitivity of the instrumental alignment which seems to have an even larger impact on the reflected channel even though the FPIs themselves are temperature stabilized to $10$ mK even during the eclipse phases.

## 7   Summary

In August 2018, ESA has launched the first Doppler wind lidar into space. In order to calibrate the instrument and to monitor

the overall instrument conditions, instrument spectral registration measurements have been performed with Aeolus on a weekly basis. During these measurements, the laser frequency is scanned over a range of $11$ GHz in order to measure the transmission curves of the spectrometers. Within this study, tools and mathematical model functions in order to analyze the measured spectrometer transmission curves were introduced and used to retrieve time series of respective fit parameters for the time





**Table 4.** Frequency drift rates and total frequency drift obtained for the direct channel and the reflected channel over two periods of operation for FM-A and FM-B as shown in Fig. 7.

| FM-A | 1st period, 84 days | 2nd period, 118 days | Total frequency drift |
|---|---|---|---|
| | 17 Oct 18 to 9 Jan 19 | 15 Feb 19 to 13 Jun 19 | |
| Dir. ch. | $(+0.26 \pm 0.03)$ MHz/d | $(+0.14 \pm 0.01)$ MHz/d | $+39$ MHz |
| Ref. ch. | $(-0.73 \pm 0.03)$ MHz/d | $(-0.15 \pm 0.02)$ MHz/d | $-75$ MHz |
| FM-B | 1st period, 221 days | 2nd period, 350 days | Total frequency drift |
| | 25 Jul 19 to 2 Mar 20 | 30 Mar 20 to 15 Mar 21 | |
| Dir. ch. | $(+0.22 \pm 0.01)$ MHz/d | $(+0.14 \pm 0.01)$ MHz/d | $+96$ MHz |
| Ref. ch. | $(+0.16 \pm 0.01)$ MHz/d | $(+0.16 \pm 0.02)$ MHz/d | $+87$ MHz |

period from October 2018 to March 2021. The models representing the FPI transmission curves is based on an Airy function
with Gaussian defects, but does also consider the spectral modification induced by the reflection on the Fizeau interferometer
which is due to the sequential spectrometer setup. Additionally, the impact of the finite aperture within the receiver and the
field of view of the illuminating beam on the FPI transmission is discussed. Based on this analysis, it is revealed that the overall
conditions were different for the respective lasers FM-A (August 2018 till June 2019) and FM-B (July 2019 till now).

The Rayleigh channel signal levels are shown to be remarkably smaller for the FM-A period. In particular, at the begin of the
respective laser period, the FM-A signal levels are smaller by about $30\%$. Furthermore it is shown that the detected signal levels
decreased over time. For the FM-A period, the decrease is rather constant for both channels (direct channel: $-0.14$ %/day,
reflected channel: $-0.13$ %/day) except for certain time periods that were related to laser parameter optimizations. As for
FM-A, the signal levels are decreasing during the FM-B time period. A laser cold plate temperature optimization performed in
March 2020 led to a signal decrease by $15\%$ but also to a significant reduction of the decrease rate. Since December 2020, the
signal levels are shown to decrease stronger than before.

The FPI center frequencies are shown to drift by several MHz per week throughout the mission. It is demonstrated that
during the FM-A period, the drift rates were rather variable and in different spectral directions for the respective FPI channel.
On the other hand, the drift was rather constant and towards similar spectral directions for both channels during the FM-B
period. This indicates that the overall illumination or rather alignment conditions were rather different for the two lasers FM-A
and FM-B. By considering the field of view of the internal reference beam ($455\,\mu$rad) as well as the finite aperture size of the
FPIs ($1.44$ mrad) it is shown that the incidence angle has to direct by several hundred $\mu$rad in order to explain the observed
center frequency drifts. Considering the angular size of the field stop at Rayleigh spectrometer level ($1.44$ mrad) and an angular
size of the Rayleigh spots ($0.968$ mrad, $4\sigma$ diameter) such a drift is rather significant and may lead to a clipping of the beam. It
is further shown that changes of the ambient temperature affect the overall instrument stability, whereas the reflected channel
is more sensitive than the direct channel. This is especially obvious in particular eclipse phases, where the satellite is partly
out of sun illumination and thus decreases its temperature. The significant observed frequency drift also explains why regular
instrument calibrations are inevitable in order to avoid systematic errors in the Aeolus wind product.



In addition to the FPI transmission curves, the characteristics of the Fizeau transmission are analyzed directly as well as from the imprint on the FPI transmission curves. It is shown that the spectral shape of the Fizeau transmission is different for the FM-A and FM-B laser, and that it evolves with time. Furthermore, for both lasers the Fizeau transmission is decreasing with time which could be explained by a shrinking beam diameter or a reduction of the beam divergence.

In the future is is foreseen to extend the presented results on Aeolus alignment drifts by using for instance the spatial information of the Rayleigh spots.

Furthermore it is pointed out that the instrumental functions and analysis tools introduced in the study may also be applied for upcoming missions using similar spectrometers as for instance EarthCARE (ESA) which is based on the Aeolus FPI design.

*Author contributions.* BW prepared the main part of the manuscript and performed the ISR analyses, CL contributed with the analysis of the ALADIN laser performance, OL provided useful information on the ALADIN laser performance, corresponding time series and laser setting changes, UM performed processor modifications and provided special data sets, OR led the presented study and helped to prepare the paper manuscript, FW provided particular data sets for the presented study, FF developed FPI mathematical models in order to determine the ALADIN alignment conditions based on ISR measurements, TF and AD contributed with discussions and helped to prepare the paper manuscript, DH developed the operational Aeolus L1B processor and provided continuous support with special processing requests, MV performed investigations regarding both the FPI and Fizeau interferometer alignment conditions and the resulting performance for Aeolus.

*Competing interests.* The authors declare that they have no conflict of interest.

*Acknowledgements.* The presented work includes preliminary data (not fully calibrated/validated and not yet publicly released) of the Aeolus mission that is part of the European Space Agency (ESA) Earth Explorer Programme. The processor development, improvement and product reprocessing preparation are performed by the Aeolus DISC (Data, Innovation and Science Cluster), which involves DLR, DoRIT, ECMWF, KNMI, CNRS, S&T, ABB and Serco, in close cooperation with the Aeolus PDGS (Payload Data Ground Segment). The analysis has been performed in the frame of the Aeolus Data Innovation and Science Cluster (Aeolus DISC).





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
