# Peer review of "Spectral performance analysis of the Aeolus Fabry-Pérot and Fizeau interferometers during the first years of operation"

_Atmospheric Measurement Techniques, 2021_

## Author Comment (AC1)

**(Reviewer comments) (Author response) (changes in the manuscript)**

As noted in lines 250-255, this is a complex system where the behavior of one interferometer impacts the downstream performance of the others. As such, this detailed description is helpful and insightful (and necessary). In fact, I would argue this is an important piece of work to have documented for posterity, particularly the mathematical descriptions of the specific Aeolus interferometer implementation and the detailed on-orbit operational performance. The mathematical approach appears correct and robust.

Doppler lidar is challenging because so many effects can appear as Doppler shifts. Changes in laser frequency, thermal shifts, plate spacing – all can appear as Doppler shifts and have to be separated from the actual atmosphere-induced Doppler shift. There is a comment early in the manuscript about thermal impacts on the telescope. Are there any thermal impacts being noted on the interferometers? Indeed, thermal impacts are not only observed on the telescope but also on the rest of the ALADIN instrument. In the bottom panel of Fig. 7, the drift of the spectral spacing (black circles) as well as the temperature measured at the ALADIN detection electronic units (DEU) is shown whereas the latter serves as a proxy for the ambient temperature within the instrument. From this figure it can be seen that the instrument temperature changes by several Kelvin throughout the annual cycle, depending on the position of the sun with respect to the satellite. For this reason, the FPIs and the polarizing beam splitter block (Fig. 1, PBSB) are placed in a thermal hood to reach a long-term temperature stability of +/- 10 mK. On a shortterm (hours), the temperature stability is better than 3 mK. Considering the temperature sensitivity of the Rayleigh channel of 455 MHz/K (81 m/s/K), 3 mK temperature fluctuations correspond to 0.2 m/s. However, as the internal reference signal is usually used for the wind retrieval, only temperature fluctuations that occur within one observation (12 s) are of relevance and are observed to be even smaller than 3 mK. In order to clarify this issue, we added the following paragraph to the manuscript:

Following line 115: "As the FPIs are rather temperature sensitive ( $\approx$  455 MHz/K which corresponds to  $\approx$  81 m/s/K), they are enclosed in a thermal hood to reach a long-term temperature stability of about ±10 mK. On the short time scale of a wind observation (12 s), the temperature stability is even better than 3 mK, which translates to wind speed variations of less than  $\approx$ 0.2 m/s".

In discussing the results shown in Figure 4 (e.g., manuscript lines 351-364), it would be helpful to know, from forward modeling, how good the fit has to be to maintain bias at <1 m/s, <5 m/s, <10 m/s, etc. In my experience, the absolute wind determination is highly sensitive to the FPI fit. Thus, even the small amount of variability as shown in Figure 5, for example, can have large impact on the wind retrieval. This manuscript does a good job of explaining and quantifying the measured instrument response. What is not answered is the question, "Is this good enough?" In other words, perhaps add a succinct description of how well the errors have to be minimized to have less than X m/s impact on the final product. This would help the reader understand how close the team is to best possible performance, or if this is best possible and is at the limit of noise.

This question is rather complex to answer, as a lot of steps are performed in the Aeolus processor to derive HLOS winds. Thus, a forward modelling of the induced systematic error is out of scope for this publication as much more detailed information about the wind processing and corresponding calibration procedures have to be addressed in advance. In particular, there is only one step in the processing chain where the FPI transmission curve fits are used, namely the so-called Rayleigh-Brillouin correction (Dabas et al., 2008, Dabas and Huber, 2017), where the impact of temperature and pressure differences in various altitudes on the receiver response is considered. To perform this correction, the fits are convolved with Rayleigh-Brillouin spectra calculated for different temperatures and pressures, and the derived response change is used for correction. However, as the atmospheric channel has different optical properties than the internal reference channel, further modifications have to be performed. In particular, the fits of the internal reference FPI transmission curves is convolved with a tilted top-hat function to consider any etendue effects. The accuracy of this procedure cannot be well assessed, as there is no possibility to measure the FPI transmission curves via the atmospheric channel with the needed accuracy. Among others, this is the reason why additional bias corrections based on ground return or ECMWF-model data are performed to obtain a wind product with a small systematic error of e.g. < 1 m/s. This bias correction is extensively explained by Weiler et al., 2021. Furthermore, as the width of the Rayleigh-Brillouin spectrum is rather broad (i.e. about 3 to 4 GHz for a laser wavelength of 355 nm and atmospheric pressures from 0 hPa to 1013 hPa and temperatures from 220 K to 330 K), the response of the Rayleigh channel is insensitive to small scale details as observed for the FPI transmission curve residuals. In order to clarify this issue, we add the following explanations to the manuscript: Following line 358: "It is also worth mentioning here, that the shown deviations cannot directly be related to a potentially origination systematic error, as several steps are performed during the wind processing chain. The only processing step that directly applies FPI transmission fit curves is the RBC that considers the impact of different atmospheric temperatures and pressures on the receiver response (Dabas et al., 2008; Dabas and Huber, 2017). Within the RBC, the FPI fit curves are convolved with Rayleigh-Brillouin spectra of different temperatures and pressures, as well as with a tilted top-hat function to consider optical differences between the internal reference and the atmospheric path. The particular accuracy of the latter procedure cannot be well assed, as there is no possibility to measure the FPI transmission curves via the atmospheric path with the needed accuracy. Among others, this is the reason why additional bias corrections based on ground return signals or ECMWF-model data are performed to obtain a wind product with a small systematic error of e.g.  $\approx 1$ m/s. This bias correction is extensively explained by Weiler et al. (2021b). Additionally, as the width of the RB spectrum is rather broad (i.e. 3 to 4 GHz for a laser wavelength of 355 nm and atmospheric temperatures and pressures), the response of the Rayleigh channel is insensitive to small scale details as observed for the FPI transmission curve residuals.". Furthermore, the following references were added:

- Dabas, A. and Huber, D.: Generation and update of AUX CSR, AE.TN.MFG-L2P-CAL-003, p. 43, https://earth.esa.int/eogateway/news/announcement-of-opportunity-for-aeolus-cal-val, 2017
- Rennie, M., Tan, D., Andersson, E., Poli, P., Dabas, A., De Kloe, J., Marseille, G.-J., and Stoffelen, A.: Aeolus Level-2B Algorithm Theoretical Basis Document (Mathematical Description of the Aeolus Level-2B Processor), ECMWF, https://earth.esa.int/eogateway/documents/20142/37627/ Aeolus-L2B-Algorithm-ATBD.pdf, 2020.

There must have been an initial expectation of controlling instrument parameters to some limits, to maintain wind error less than some specified amount. How well did the initial expectation or modeling match the measured results?

The specifications for the spectrometer parameters that are available are indicated in Table 1. For an easier comparison with the determined parameters from the ISR measurements, we added the corresponding specifications including their margins to Table 3, which shows the fit results of one of the first ISR measurements that was performed on 10 October 2018.

| Parameter                                                                                                                                                                                                                                                                              | Unit | Dir. ch. $\mathcal{T}_{dir}(f)$         | Ref. ch. $\mathcal{T}_{ref}(f)$       | Specification                                   |
|----------------------------------------------------------------------------------------------------------------------------------------------------------------------------------------------------------------------------------------------------------------------------------------|------|-----------------------------------------|---------------------------------------|-------------------------------------------------|
| I                                                                                                                                                                                                                                                                                      | LSB  | $3765 \pm 3 (3722 \pm 4)$               | $3209 \pm 2 \ (3120 \pm 2)$           | -                                               |
| $\mathcal{I}$ ratio (integral)                                                                                                                                                                                                                                                         | -    | 0.85 (0.84)                             |                                       | -                                               |
| R                                                                                                                                                                                                                                                                                      | -    | $0.649 \pm 0.001 \ (0.651 \pm 0.001)$   | $0.653 \pm 0.001 \ (0.652 \pm 0.001)$ | $0.65 \pm 1\%^{***}$                            |
| $\sigma_{g}$                                                                                                                                                                                                                                                                           | MHz  | $138 \pm 6 \ (147 \pm 7)$               | $156 \pm 6 \ (147 \pm 7)$             | -                                               |
| $f_0$                                                                                                                                                                                                                                                                                  | GHz  | -1.236 ( $-1.239$ )                     | 4.220 (4.217)                         | -                                               |
| Spacing                                                                                                                                                                                                                                                                                | MHz  | 5456 (5456)*                            | 5490 (5490)                           | $5477 \pm 120 \ (2.3 \pm 0.05 \text{ pm})^{**}$ |
| $\Gamma_{FSR}$                                                                                                                                                                                                                                                                         | MHz  | 10946 (fixed)                           | 10946 (fixed)                         | $10955 \pm 48 \ (4.6 \pm 0.02  \text{pm})^{**}$ |
| Q                                                                                                                                                                                                                                                                                      | -    | -                                       | $0.93 \pm 0.01 \ (0.92 \pm 0.01)$     |                                                 |
| $\mathcal{I}_{\mathrm{Fiz}}$                                                                                                                                                                                                                                                           | -    | $0.144 \pm 0.003$ ( $0.141 \pm 0.006$ ) | $0.136 \pm 0.003 \ (0.141 \pm 0.004)$ | -                                               |
| $f_{0_{\rm Fiz}}$                                                                                                                                                                                                                                                                      | GHz  | -2.689(-2.691)                          | -2.582(-2.573)                        | -                                               |
| $\Gamma_{\rm FSR_{Fiz}}$                                                                                                                                                                                                                                                               | MHz  | $2202 \pm 8 \ (2205 \pm 6)$             | $2177 \pm 6 \ (2175 \pm 3)$           | $2191 \pm 24 \ (0.92 \pm 0.01 \text{ pm})^{**}$ |
| $d_{\rm Fiz}$                                                                                                                                                                                                                                                                          | -    | 0.5 (fixed)                             | 0.5 (fixed)                           | -                                               |
| $FWHM_{ref}$                                                                                                                                                                                                                                                                           | MHz  | $1519 \pm 4 \ (1508 \pm 3)$             | $1496 \pm 3 \ (1500 \pm 3)$           | 1516***                                         |
| $FWHM_{def}$                                                                                                                                                                                                                                                                           | MHz  | $325 \pm 14 \ (347 \pm 16)$             | $368 \pm 15 \ (345 \pm 17)$           | 695***                                          |
| $FWHM_{tot}$                                                                                                                                                                                                                                                                           | MHz  | $1591 \pm 4 \ (1589 \pm 6)$             | $1587 \pm 5 \ (1580 \pm 6)$           | $1667 \pm 7 \ (0.70 \pm 0.003 \text{ pm})^{**}$ |
| Finesse tot                                                                                                                                                                                                                                                                 | -    | $6.88 \pm 0.02$ (6.89 $\pm 0.02$ )      | $6.90 \pm 0.02 \ (6.93 \pm 0.02)$     | 6.6**                                           |
| The fit values in brackets denote the one retrieved from the energy drift corrected data set (see also Sect. 3.2).
* Direct channel is at lower frequencies. The second cross point is calculated by considering the measured FSR.
** Values as eiven by Reitebuch et al. (2009) |      |                                         |                                       |                                                 |

Table 3. Fit parameters according to Eq. (9), Eq. (11) and Eq. (12). The given uncertainty of the fit values denotes the standard error derived by the fit routine.

\*\*\* Values taken from internal documents of the manufacture that are not available for public. n/a = no specification available.

Additionally, a comparison of the determined parameters and the corresponding specifications is given at the respective position within the manuscript:

- In line 330 we modified the sentence according to "The given uncertainty of the fit values denotes the standard error derived by the fit routine and the last column indicates the specification values for comparison".
- Following 342 we add the following sentence: "Furthermore is can be seen that the derived spectral spacing on both sides of the filter curves is well within the specification of (5477 ± 120) MHz."
- Following 346 we add the following sentence: "...when neglecting the imprint of the reflection on the Fizeau interferometer and are smaller than the specified value of (1.667 ± 7) GHz.".

Minor point: Figure 4 is referenced in line 183, well before it appears in the document and before Figure 3. Best practice is to have the figures in the order they are referenced.

We removed the ling to Fig. 4 here and just referenced the section were this topic is further discussed. The sentence now reads: "This is especially obvious from the tilt that is visible in the relative residuals as discussed later in Sect. 4.3.".

Grammatical issue: The authors use the phrase "in order to" nearly 40 times in this manuscript. Use of "in order to" is poor grammar. It's one of the few useful things I retain from grammar school, but "in order to" should never be used. Instead, just "to." For example, "In order to provide valuable input data..." is more appropriately simply "To provide valuable input data..."

We revised the manuscript regarding your suggestion and replaced "in order to" throughout the manuscript.

---

## Author Comment (AC2)

**(Reviewer comments) (Author response) (changes in the manuscript)**

1. The optical frequency discrimination performance of Doppler lidar seriously affects the accuracy of wind speed measurement. This paper does not mention how the frequency discrimination performance affects the measurement accuracy with the system operation.
This question is rather complex to answer, as a lot of steps are performed in the Aeolus processor to derive HLOS winds. Thus, it is difficult or rather impossible to relate any FPI alignment drifts directly to a systematic error of the derived wind speed. Hence, also a forward modelling of the induced systematic error is out of scope for this publication, as much more detailed information about the wind processing and corresponding calibration procedures have to be addressed in advance. In particular, there is only one step in the processing chain where the FPI transmission curve fits are used, namely the so-called Rayleigh-Brillouin correction (Dabas et al., 2008, Dabas and Huber, 2017), where the impact of temperature and pressure differences in various altitudes on the receiver response is considered. To perform this correction, the fits are convolved with Rayleigh-Brillouin spectra that are calculated for different temperatures and pressures, and the derived response change is used for correction. However, as the atmospheric channel has different optical properties than the internal reference channel, further modifications have to be performed. In particular, the fits of the internal reference FPI transmission curves are convolved with a tilted top-hat function to consider any optical etendue effects, which considers the different optical illuminations of the atmospheric and reference channel. The accuracy of this procedure cannot be well assessed, as there is no possibility to measure the FPI transmission curves via the atmospheric channel with the needed accuracy. Among others, this is the reason why additional bias corrections based on ground return or ECMWF-model data are performed to obtain a wind product with a small systematic error of e.g. < 1 m/s. This bias correction is extensively explained by Weiler et al., 2021. Furthermore, as the width (FWHM) of the Rayleigh-Brillouin spectrum is rather broad (i.e. about 3 to 4 GHz for a laser wavelength of 355 nm and atmospheric pressures from 0 hPa to 1013 hPa and temperatures from 220 K to 330 K), the response of the Rayleigh channel is insensitive to small scale details as observed for the FPI transmission curve residuals. In order to clarify this issue, we add the following explanations to the manuscript: Following line 358: "It is also worth mentioning here, that the shown deviations cannot directly be related to a potential systematic error, as several steps are performed during the wind processing chain. The only processing step that directly applies FPI transmission fit curves is the RBC that considers the impact of different atmospheric temperatures and pressures on the receiver response (Dabas et al., 2008; Dabas and Huber, 2017). Within the RBC, the FPI fit curves are convolved with Rayleigh-Brillouin spectra of different temperatures and pressures, as well as with a tilted top-hat function to consider the different optical illumination between the internal reference and the atmospheric path. The particular accuracy of the latter procedure cannot be well assed, as there is no possibility to measure the FPI transmission curves via the atmospheric path with the needed accuracy. Among others, this is the reason why additional bias corrections based on ground return signals or ECMWF-model data are performed to obtain a wind product with a small systematic error of below e.g. ≈ 1 m/s. This bias correction

is extensively explained by Weiler et al. (2021b). Additionally, as the width of the RB spectrum is rather broad (i.e. 3 to 4 GHz for a laser wavelength of 355 nm and atmospheric temperatures and pressures), the response of the Rayleigh channel is insensitive to small scale details as observed for the FPI transmission curve residuals.". Furthermore, the following references were added:

- Dabas, A. and Huber, D.: Generation and update of AUX CSR, AE.TN.MFG-L2P-CAL-003, p. 43, https://earth.esa.int/eogateway/news/announcement-of-opportunity-for-aeolus-cal-val, 2017
- Rennie, M., Tan, D., Andersson, E., Poli, P., Dabas, A., De Kloe, J., Marseille, G.-J., and Stoffelen, A.: Aeolus Level-2B Algorithm Theoretical Basis Document (Mathematical Description of the Aeolus Level-2B Processor), ECMWF, https://earth.esa.int/eogateway/documents/20142/37627/Aeolus-L2B-Algorithm-ATBD.pdf, 2020.

2. This paper mentioned that the frequency discrimination performance of Doppler lidar is mainly caused by the alignment stability and the laser quality. In my experience, the performance of the detector also affects the efficiency of photoelectric conversion, especially for ACCD.
It is agreed that the quantum efficiency of the detector affects the overall signal level and with that the magnitude of the random error. However, regarding the instrumental alignment and the spectrometer performance, the ACCD detectors play a minor or even negligible role. The quantum efficiency of the ACCD detectors was characterized to be 85% (Reitebuch et al., 2009, Weiler et al., 2021), and there are no measurements available that indicate any change of the quantum efficiency during the mission lift-time. An issue that was identified quite at the beginning of the mission but not during ground tests was, that the ACCD detectors suffer from an increasing number of hot-pixels that arise during life time and that can induce a systematic error in the retrieved wind speeds. Thus, a special measurement mode as well as a corresponding correction procedure was developed as described by Weiler et al., 2021a (see also line 37). However, these hot-pixels arise only in the memory zone of the detector and thus, do not affect ISR measurements with the internal reference signal.

3. Why don't chose PMT detectors in this system, and apply FPI for Doppler frequency discrimination of Mie doppler signals?
Although the original system specifications are not in the scope of this paper, it is tried to explain the reasons why the actual Aeolus spectrometers and detectors were chosen with the current specifications. Since the spectral widths of the molecular backscattered light (FWHM ~ 3-4 GHz) and the light backscattered on particles (FWHM ~0.05 GHz) differ by almost 2 orders of magnitude, different spectrometers with different bandwidths have to be used for frequency discrimination.
For the Mie channel, it was decided to use a Fizeau interferometer in fringe-imaging configuration to analyze the narrowband return of particulate scattered light. Using a Fizeau interferometer is beneficial here, as the formed interference pattern is a "straight" line (fringe) that can easily be imaged onto a 2D-detector and can be analyzed more easily compared to the circular interference pattern that would originate from an FPI. In addition, the ACCD is able to accumulate returns from successive laser pulses directly on the chip in an internal accumulation register. Thus, the summation of pulse returns does not need a reading of each one of them, which would lead to additional readout noise.
For the Rayleigh channel, it was decided to use the common and approved double-edge technique. To prevent any additional efforts with additional space-qualification procedures of additional detectors (e.g. PMTs), it was decided to use the same ACCD detector for both channels. Furthermore, the ACCDs provide a rather good quantum efficiency of 85% at ultraviolet wavelengths, and low dark current and read-out noise, which allows quasi-photon counting. In addition, having a 2D ACCD detector for the Rayleigh channel gives further opportunities to analyze the system performance. For instance, the Rayleigh spot position behind/after the FPIs provides further information on the system alignment and potentially ongoing drifts.

4. How to normalize the long-term FPI transmission curves, in the line 380 and why the direct channel normalization forms of Fig. 5(a) and Fig. 5(b) are inconsistent?

The direct channel transmission curves are normalized to unit area, and the reflected channel is scaled with the same value as the direct channel in order to keep the ratio between the two transmission curves. Furthermore, the x-axis is normalized to the direct channel center frequency, whereas the center frequency is chosen from the fit. The normalization of the direct channel transmission curves for the FM-A period (Fig. 5, a) and FM-B period (Fig. 5, b) is consistent but looks different. This has mainly to do with the changing modulations that are caused by the reflection on the Fizeau interferometer and a changing laser beam profile. These modulations were larger for the FM-B period (e.g. at +1.0 GHz) and lead to varying peak intensity levels when the normalization is performed to unit area. To clarify the normalization, the following explanation was added: "To be able to directly compare respective measurements, the direct channel transmission curves are normalized to unit area such that $\int_0^{\mathcal{F}_{\mathrm{FSR}}} \mathcal{T}_{\mathrm{dir}}(f)\,df = 1$, and the reflected channel is normalized accordingly with the same factor as used for the direct channel to keep the ratio between the two respective channels.".

5. The beam divergence angle of the incident FPI has a serious impact on the transmittance of the FPI in line 570. The larger divergence angle corresponds to the greater FWHM and the lower transmittance of the FPI. According to the optical path setting in Fig. 1, only one collimating mirror is used to collimate the received signal light. The signal light is reflected by the Fizeau interferometer and then enters the direct channel of the FPI. The reflected light of the direct channel enters the reflection channel of the FPI, and the optical path is too long. How to ensure the incident FPI with minimal divergence angle.

In line 83 of the original manuscript, we provide the following information: "Furthermore, the transmit-receive optics contain a field stop (FS) with a diameter of about 88 μm in order to set the field of view (FOV) of the receiver to be only 18 μrad which is needed to limit the influence of the solar background radiation and the incidence angle on the spectrometers.". Thus, the instrument field of view and the spectrometer divergence is basically determined by the size of the filed stop. As the input beam diameter changes for the Fizeau interferometer and the FPIs, and as the étendue is preserved, the spectrometers are illuminated with a different divergence, but the divergence is the same for the direct channel and the reflected channel FPI. In principle, the divergence on the spectrometers could be further reduced by minimizing the size of the field stop or increasing the beam size and aperture of the interferometer. However, this would require a further reduced laser beam divergence to avoid clipping and lead to higher costs. Furthermore, the system would be more sensitive to alignment changes in case of an even smaller field stop size.

6. In line 135, 20 laser pulses for each measurement are used, while 190 pulses is used in 145 line, please explain it clearly.

During an ISR measurement, three different frequency steps are performed per observation, and each observation consist of 30 measurements with 19 laser pulses. Thus, one frequency step lasts for 10 measurements and contains the data of 190 laser pulses. To clarify this issue, the following sentence was added or rather extended: Following line 139: "…83790 laser pulses, and each measurement at a certain frequency step consists of 10 measurements and contains the data of 190 laser pulses" Additionally, we mentioned in line 138 that every last laser pulse of a measurement does not contribute to the accumulated signal, thus using only 19 out of 20 pulses.

7. Lines 155-190 describe the process of laser energy drift correction. For the reader's convenience, a simple system schematic diagram including optical and electrical devices is recommended.

It is agreed that the applied laser energy drift correction was not explained sufficiently in the original manuscript. To further clarify, we added an additional figure and a paragraph: Following line 189: "... factor that is derived by the analysis of FPI transmission curve residuals such that the residual exhibits no skewness anymore. This procedure is illustrated in Fig. 3, where panel (a) shows the laser energy measured by PD-74 and normalized to the first data point $E_{norm}(f)$ (blue) as well as the corrected normalized laser energy $E_{normnew}(f)$ (orange) for the ISR measurement performed on 10 October 2018. In panel (b), the respective FPI transmission curves of the direct channel, derived by using $E_{norm}(f)$ (blue) and $E_{normnew}(f)$ (orange) for the laser energy drift correction are shown. The corresponding relative residuals are shown in panel (c). The line fits applied to the data reveals that the slope is close to zero, when $E_{normnew}(f)$ is used for the laser energy correction (orange), whereas a significant skewness is obvious when $E_{norm}(f)$ is used (blue). It can be seen that the relative deviations vary between -2% and 4% (peak-to-peak), whereas the distinct modulation is caused by an insufficient description of the spectral features of the Fizeau reflection and modulations of the incident laser beam profile and/or the transmission over the Fizeau aperture (see also Fig. 4 and the corresponding discussion). For the ISR measurement on 10 October 2018, $\xi_i$ was determined to be 0.04. For the entire mission time discussed in this paper (October 2018 until March~2021), $\xi_i$ varies between 0.05 and -0.18. ".

[Figure]

**Figure 3.** (a): Normalized laser energy $E_{norm}(f)$ (blue) and corrected normalized energy $E_{norm_{new}}(f)$ (orange) versus commanded laser frequency for the ISR measurement performed on 10 October 2018. (b): Corresponding FPI transmission curves of the direct channel according to Eq. (2) by using $E_{norm}(f)$ (blue dots) or rather $E_{norm_{new}}(f)$ (orange dots) for the laser energy drift correction. To illustrate the small differences, the y-axes is plotted with logarithmic scale. (c): Relative residuals of the best fits according to Eq. (9) and line-fits.